# Neuronal connectome of a sensory-motor circuit for visual navigation

**Nadine Randel, Albina Asadulina, Luis A Bezares-Calderón, Csaba Verasztó, Elizabeth A Williams, Markus Conzelmann, Réza Shahidi, Gáspár Jékely\***

Max Planck Institute for Developmental Biology, Tübingen, Germany

**Abstract** Animals use spatial differences in environmental light levels for visual navigation; however, how light inputs are translated into coordinated motor outputs remains poorly understood. Here we reconstruct the neuronal connectome of a four-eye visual circuit in the larva of the annelid *Platynereis* using serial-section transmission electron microscopy. In this 71-neuron circuit, photoreceptors connect via three layers of interneurons to motorneurons, which innervate trunk muscles. By combining eye ablations with behavioral experiments, we show that the circuit compares light on either side of the body and stimulates body bending upon left-right light imbalance during visual phototaxis. We also identified an interneuron motif that enhances sensitivity to different light intensity contrasts. The *Platynereis* eye circuit has the hallmarks of a visual system, including spatial light detection and contrast modulation, illustrating how image-forming eyes may have evolved via intermediate stages contrasting only a light and a dark field during a simple visual task.

## Introduction

Visually guided behavior is widespread in animals (*Ullén et al., 1997*; *Garm et al., 2007*; *Orger et al., 2008*; *Burgess et al., 2010*; *Huang et al., 2013*), yet the underlying neuronal circuits and their evolutionary origins remain poorly understood. Spatial vision requires at least two photoreceptors and a neural circuitry capable of making a comparison between the photoreceptor inputs without body movement (*Land and Nilsson, 2002*; *Nilsson, 2009*). The spatial information obtained must then translate to a coordinated motor output.

A comprehensive description of the sensory-motor visual circuitry, including all neurons and their synaptic connectivity, is required for a plausible explanation of how visual inputs drive motor output during animal behavior. This can only be achieved using electron microscopic imaging to construct connectomes, comprehensive synaptic-level connectivity maps for large blocks of neural tissue containing behaviorally relevant circuits (*Bock et al., 2011*; *Briggman et al., 2011*; *Jarrell et al., 2012*; *Bumbarger et al., 2013*; *Helmstaedter et al., 2013*). However, despite recent advances in the connectomics of visual systems (*Briggman et al., 2011*; *Rivera-Alba et al., 2011*; *Sprecher et al., 2011*; *Takemura et al., 2011*, *2013*), a complete synaptic-level connectivity-map of a visual circuit, including sensory-, inter-, and motorneurons, has not yet been described.

Here we reconstruct the neural connectome of the visual eyes in a larva of the marine annelid *Platynereis dumerilii* using serial-section transmission electron microscopy (ssTEM). *Platynereis* larvae develop four visual eyes, the 'adult eyes' (henceforth 'eyes'), that are the precursors of the adult's visual pigment-cup eyes, and are distinct from the more ventrally located 'eyespots' (*Jékely et al., 2008*). These eyes consist of only 2–7 photoreceptors, a few shading pigment cells, and a lens, representing the simplest visual eyes described to date (*Rhode, 1992*; *Arendt et al., 2002*; *Randel et al., 2013*). The *Platynereis* larval visual connectome consists of 71 neurons and 1106 synapses, and was reconstructed from a tissue block containing the larval head and trunk. Using behavioral experiments combined with eye ablations we demonstrate that the eyes mediate spatial vision, whereby light

**\*For correspondence:** gaspar.
jekely@tuebingen.mpg.de

**Competing interests:** The authors declare that no competing interests exist.

**Reviewing editor**: Eve Marder, Brandeis University, United States

**eLife digest** Many animals show automatic responses to light, from moths, which are attracted to light sources, to cockroaches, which are repelled by them. This phenomenon, known as phototaxis, is thought to help animals navigate through their environment. It is an evolutionarily ancient behavior, as revealed by its widespread presence in the animal kingdom. One animal with a simple visual system for phototactic behavior is the marine worm *Platynereis dumerilii*.

Platynereis is a segmented worm (annelid) with four eyes on the top of its head, two on the right and two on the left. Exposure to light triggers the contraction of muscles that run along the length of the body, causing the worm to bend and thus change the direction it is swimming in. Now, using a combination of high-resolution microscopy and behavioral experiments in larvae, Randel et al. have mapped the neural circuits underlying the worm's phototactic behavior.

A 3-day-old Platynereis larva was sectioned to produce almost 1700 slices, each less than 50 nanometers thick, which were then viewed under a transmission electron microscope. By tracing individual neurons from one slice to the next, it was possible to reconstruct the entire visual system and all of its connections. This 'visual connectome' consisted of 71 neurons—21 light-sensitive cells, 42 interneurons, and 8 muscle-controlling motorneurons—organized into a circuit with 1106 connections.

Shining light onto living larvae triggered phototaxis, with some larvae consistently swimming towards the light and others away from it. Using a laser to destroy all four eyes abolished this behavior, as did the removal of both eyes on either side of the head. By contrast, removing one eye from each side had no effect. This was because these larvae were still able to simultaneously compare the amounts of light reaching the left and right sides of their body, and to use any difference in these levels as a directional cue to guide swimming.

By revealing the circuitry underlying phototaxis in a marine worm, Randel et al. have provided clues to the mechanisms that support this behavior in other species. The data could also provide insights into the processes that contributed to the evolution of more complex visual systems.

intensities at the left and right eyes are compared to mediate tail bending during phototactic turns. This combination of connectomics and behavioral analysis provides a circuit-level and mechanistic explanation for the regulation of phototactic behavior by spatial vision. The *Platynereis* visual connectome also provides insights into the origin of visual eyes, and suggests that phototaxis may have been the first visual task in evolution performed by animals.

## Results

### Eye-circuit connectome reconstruction

To reconstruct the full chemical synaptic connectivity map of the visual eyes in the *Platynereis* larva we used ssTEM. Following high-pressure freezing and cryosubstitution, we collected 1690 thin sections (40–50 nm) from a single Epon-embedded 3-day-old larva, encompassing the entire head and part of the first trunk segment. Following contrasting, we imaged the sections at 3.7 nm/pixel resolution, in a 140 µm × 140 µm × 80 µm volume (*Figure 1A–C*; *Video 1*).

We identified the eyes based on the presence of pigment-filled vesicles in the pigment cells, the presence of a lens formed by the apical extensions of the pigment cells, and the presence of the apical microvillar extensions (rhabdoms) of the photoreceptors (*Rhode, 1992*; *Randel et al., 2013*; *Figure 1D*; *Video 2*). We then manually traced all neurons lying on synaptic pathways from photoreceptors to locomotor organs (muscles and ciliary bands). We identified 21 photoreceptor cells (PRC), 42 interneurons (IN) and 8 motorneurons (MN) (*Figure 1—figure supplement 1–4*), forming a 'minimal eye circuit' from sensors to effectors. We identified 1106 chemical synapses between these cells and their motor targets. We found further sensory neurons other than the photoreceptors providing direct or indirect input to the minimal eye circuit. These upstream circuits will be described in detail elsewhere. In this paper we focus on the analysis of the 71 neurons constituting the minimal eye circuit.

First, we describe the anatomy and synaptic contacts of the eyes and the main neuron types identified as part of the eye circuit. To provide an easily accessible representation of the data we generated

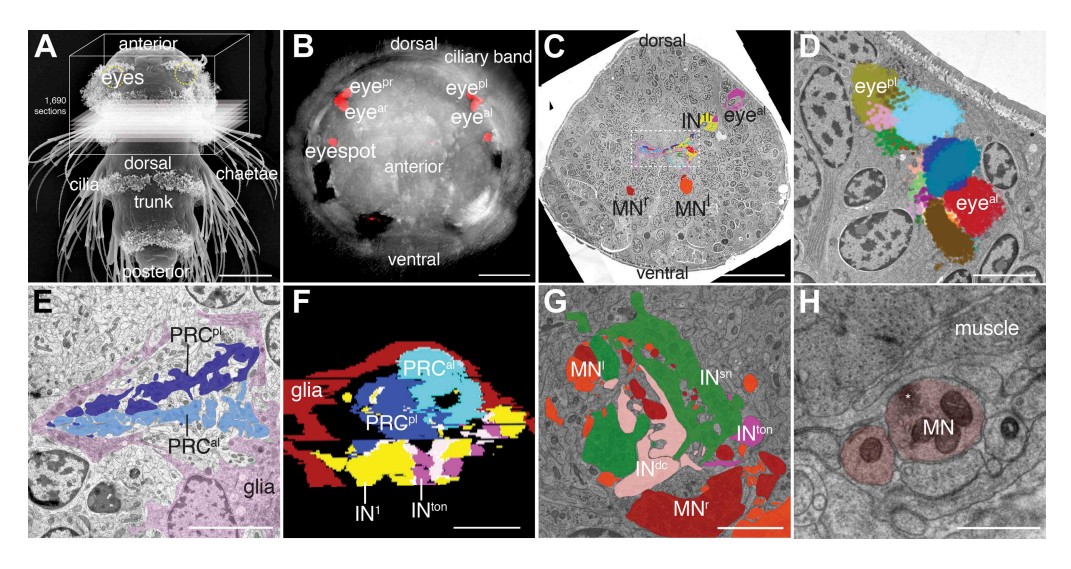

**Figure 1**. Serial-section electron microscopy imaging of the visual eye circuit in a *Platynereis* larva. (**A**) Scanning electron micrograph of a 72 hr-post-fertilization larva, dorsal view. The boxed volume was sectioned and imaged. (**B**) Anterior view of a 72 hr-post-fertilization larva visualized with differential interference contrast (DIC) optics (grey) showing the position of the eyes visualized by the reflection of the pigments (red). (**C**) A representative electron micrograph from the series with traced neurons. The boxed area contains the primary optic neuropil. (**D**) Reconstruction of the pigment cup of the left anterior and posterior eyes. Pigment granules from the different pigment cells and photoreceptors are colored differently. (**E**) TEM image of the left primary optic neuropil surrounded by glia (pink), with the photoreceptor projections from the anterior and posterior eye colored differently. (**F**) A virtual cross-section of the primary optic neuropil based on ssTEM shows the anterior-posterior layering of glia, photoreceptor, primary interneuron (IN[1]) and trans-optic-neuropil interneuron (IN[ton]) processes, anterior is up. (**G**) TEM image of the secondary optic neuropil, with segmented IN[ton], IN[sn], IN[dc] and ipsi- and contralateral motorneuron projections. (**H**) TEM image of a neuromuscular synapse from a motorneuron to the ventral longitudinal muscle. Asterisk marks a cluster of synaptic vesicles. Eye[al], anterior-left eye; eye[ar], anterior-right eye; eye[pl], posterior-left eye; eye[pr], posterior-right eye; PRC, photoreceptor; IN, interneuron; MN, motorneuron. Scale bars, 50 µm (**A**), 30 µm (**B** and **C**), 5 µm (**D**, **E**, **G**), 1 µm (**F**), 0.5 µm (**H**).

The following figure supplements are available for figure 1:

**Figure supplement 1**. Morphology of photoreceptor cells reconstructed from serial TEM sections.

**Figure supplement 2**. Morphology of IN1, IN[ton], IN[sn], and IN[dc] cells reconstructed from serial TEM sections.

**Figure supplement 3**. Morphology of IN[int] and IN[vc] cells reconstructed from serial TEM sections.

**Figure supplement 4**. Morphology of motorneurons reconstructed from serial TEM sections.

**Figure supplement 5**. Axon diameter and synapse size in the *Platynereis* larval connectome.

**Figure supplement 6**. Synapses of photoreceptors and IN[1] cells.

**Figure supplement 7**. Synapses of IN[1] cells.

**Figure supplement 8**. Synapses of IN[ton] cells and IN[sn] cells.

**Figure supplement 9**. Synapses of motorneurons.

a comprehensive 3D anatomical atlas of all neurons, synapses, muscles and ciliated cells using Blender (http://www.blender.org/) (*Randel et al., 2014*; *Video 3*). Neuron morphologies were reconstructed based on the raw traces (*Figure 1—figure supplement 1–4*) and represented in the atlas with a simplified

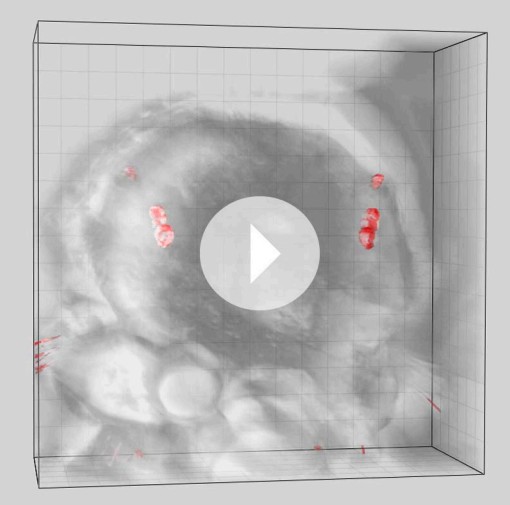

**Video 1**. Head and first trunk segment of a 3-day-old Platynereis larva. DIC and reflection imaging of the head and first trunk segment of a 73 hr-post-fertilization *Platynereis* larva, corresponding to the volume analyzed by ssTEM. The four eyes and the eyespots are visualized based on the reflection of the pigment (red). The frame contains a volume of 151 × 151 × 61 μm. The larva is squeezed dorso-ventrally with the coverslip. Grid spacing is 10 μm.

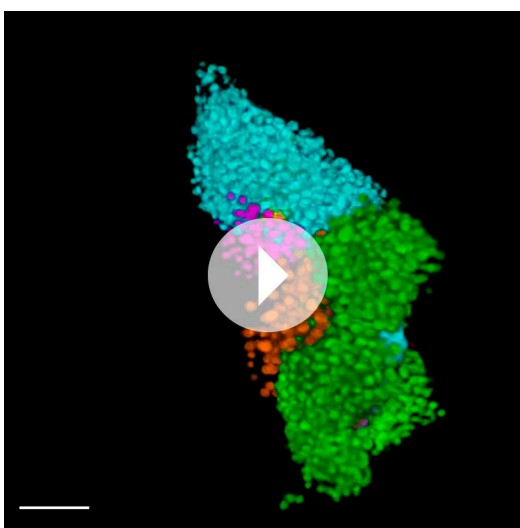

**Video 2**. Volume reconstruction of the two left eyecups. The pigment vesicles of the pigment cells and the rhabdoms of the photoreceptors were reconstructed by ssTEM. The pigment of the photoreceptors is on the convex surface of the pigment cup, shown in different colors. The photoreceptor rhabdoms are inside the pigment cup. Scale bar, 2 μm.

morphology (*Figure 2*). Users can display groups of neurons, query for pre- and post-synaptic partners of neurons, and display synapses.

## Morphology of neurons and synapses

The neurons of the eye circuit only contain axo-axonal synapses. With the exception of the photoreceptors, all neurons lack dendrites. Axons have a median diameter of 160 nm (*Figure 1—figure supplement 5*), allowing reliable tracing of most processes. The morphology of all neurons is very simple, with one primary branch giving rise to several very short secondary branches in the synaptic regions of the optic neuropils (*Figure 1—figure supplement 1–4*). These short branches often contain pre- or postsynaptic sites (*Video 4*). Only two neurons have a branched main axon (MN[r3], MN[l1], see 'Materials and methods' for nomenclature).

Synapses were identified as electron dense accumulations of several vesicles close to the presynaptic membrane (no T-bars as in *Drosophila*). Prominent postsynaptic structures were not detected, in agreement with previous ultrastructural studies in annelids (*Wells et al., 1972*). 69% of the synapses could be identified in at least two and up to five consecutive sections (*Figure 1—figure supplement 5*). To further characterize the ultrastructure of synapses we created high-resolution images (0.2 nm/pixel) of 60 randomly chosen synapses belonging to the different neuron types (*Figure 1—figure supplement 6–9*). In all cases we observed a cluster of vesicles adjacent to the plasmamembrane, but no pre- or postsynaptic specializations.

We could not identify gap junctions in the eye circuit. Gap junctions exist in annelids, and we could find several innexin genes, encoding gap junction proteins of invertebrates (*Kandarian et al., 2012*), in the *Platynereis* transcriptome (data not shown). However, we did not see structures similar to ultrastructurally characterized annelid gap junctions (*Muller and Carbonetto, 1979*; *Shen et al., 2002*), even in a stack of high-resolution (1.13 nm/pixel) images of the primary optic neuropil (*Randel et al., 2014*). If gap junctions are present in the larval stage they may be too small to be distinguished from obliquely cut membranes even at high resolution.

## Synaptic contacts in the eye circuit

The photoreceptors of the four eyes (3–7 per eye) project to a primary optic neuropil area at the center of the larval brain (*Figure 1C,E*, *Figure 2B*), consistent with results obtained by the genetic

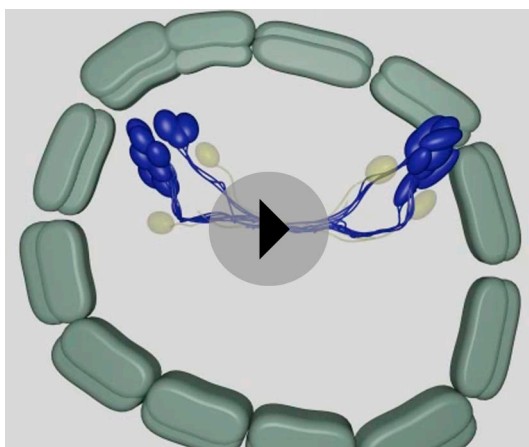

**Video 3**. Cellular complement of the Platynereis larval visual circuit. 3D reconstruction of the cellular complement of the *Platynereis* larval visual circuit. The different cell types appear in the following order: photoreceptors, IN[1], IN[int], IN[ton], IN[sn], IN[dc], IN[vc], motorneurons, synapses, glia, muscles.

labeling of the photoreceptors (*Backfisch et al., 2013*). All photoreceptors from the same eye project along the same axon bundle, forming four separate eye nerves (*Figure 2A,B*).

The eye nerves from the anterior and posterior eyes from the same side innervate distinct anterior and posterior areas in the primary optic neuropil (*Figure 1E,F*). Some photoreceptor axons cross the midline and project to the contralateral primary optic neuropil. The primary optic neuropil is surrounded by three giant glia cells. These cells form lamellae that tightly surround the primary and secondary optic neuropils from the dorsal but not the ventral side (*Figure 1F*, *Figure 2C*; *Video 4*). The main targets of the photoreceptors of the four eyes are four primary interneurons (IN[1]), with photoreceptors from one eye forming several (up to 30) synaptic contacts to one primary interneuron with a contralateral cell body (e.g., PRC[al] to IN[1pr]; *Video 4*). The four eyes and the four primary interneurons show a crosswise arrangement (*Figure 2D*). Primary interneurons have mutual synaptic contacts (see below) and also form synapses on two other interneuron types (*Figure 2E,H*; *Video 3*; *Randel et al., 2014*). One type of interneuron is intrinsic to the optic neuropil (IN[int]) with both ipsilateral and contralateral cell bodies and axons crossing the midline. Another target of the primary interneurons are interneurons that project out of the primary optic neuropil into a secondary optic neuropil area (*Figure 1G*). The major targets of these trans-optic-neuropil interneurons (IN[ton]) are a group of contralateral interneurons, which we named Schnörkel interneurons (IN[sn]). Schnörkel interneurons project a curved axon to the ipsilateral secondary optic neuropil (*Figure 2H*, *Figure 2—figure supplement 1*). The Schnörkel interneurons are presynaptic to a bilateral group of ventral motorneurons (MN; *Figure 2—figure supplement 1*). Two groups of dorsal and ventral interneurons (IN[dc], IN[vc]) also form synapses on the Schnörkel interneurons and motorneurons (*Figure 2H*; *Video 3*).

We identified two distinct motorneuron types. The first type sends a contralateral projection to both the ciliary band and the dorsal longitudinal muscle, and branches to send a descending projection to the trunk (*Figure 2G*, *Figure 2—figure supplement 1*). The second type only projects posteriorly along the circumesophageal nerve after crossing the midline. Both motorneuron types form neuromuscular and neurociliary synapses (*Figure 1H*, *Figure 2—figure supplement 1*; *Video 3*, *Video 5*). Due to the lack of further trunk sections we did not trace descending motorneuron axons along their entire length, therefore we cannot exclude the possibility that motorneurons have other synaptic partners in the trunk. Nevertheless, these data represent, to our knowledge, the most complete connectomic reconstruction of a visual circuit to date, from photoreceptors to effector muscles and ciliated cells.

## Synapse types in the eye circuitry

To gain insights into the nature of synapses in the eye circuit we measured vesicle diameter in the high-resolution images of synapses (*Figure 3—figure supplement 1*). The mean diameter of synaptic vesicles in interneurons was significantly larger than in photoreceptors and motorneurons. Photoreceptor synaptic vesicles were not significantly different from neuromuscular synaptic vesicles in motorneurons. However, neurociliary synaptic vesicles in motorneurons were significantly smaller than photoreceptor or neuromuscular synaptic vesicles. These observations indicate the use of different neurotransmitters in the eye circuitry and also suggest that the dual-function muscle- and cilio-motor neurons (MN[r2], MN[l1]) may have mixed neurotransmitter content.

To identify possible neurotransmitters in the eye circuit we performed whole-mount RNA in situ hybridizations with marker genes for various neurotransmitters (*Randel et al., 2014*) and compared the expression domains to cell body positions in the TEM series (*Figure 3*). In *Platynereis* larvae the colocalization of distinct gene expression patterns can be determined with near cellular resolution

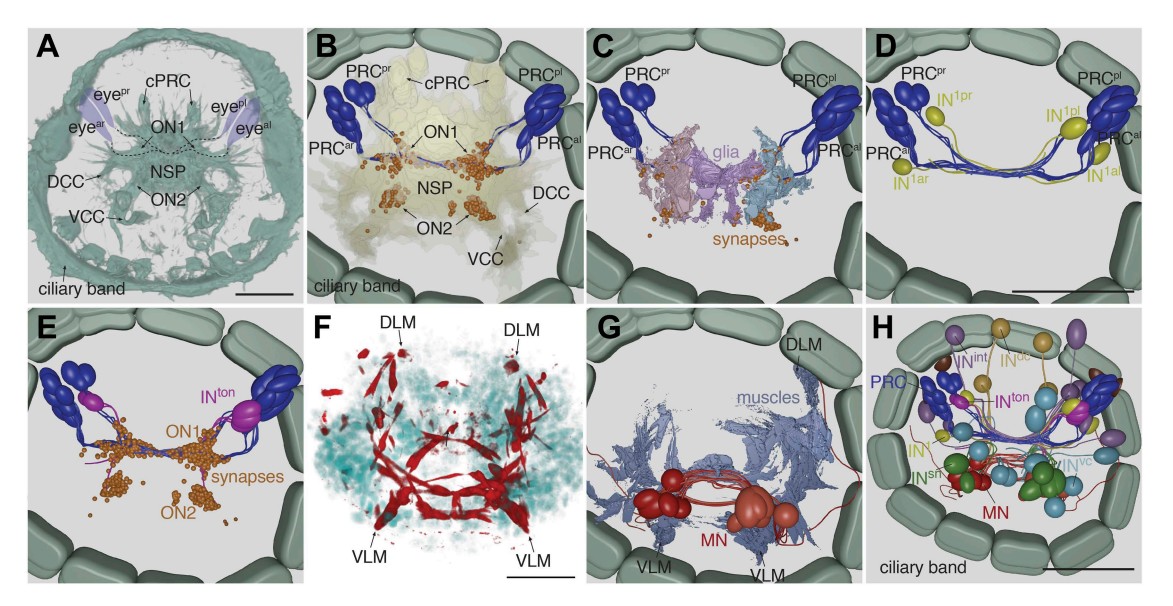

**Figure 2**. Cell complement of the visual circuit. (**A**) Confocal microscopic image of a 3-day-old larva stained with an anti-acetylated tubulin antibody to label neurites and cilia. Anatomical landmarks are labeled. (**B**) Blender visualization of all photoreceptor cells and all synapses of the minimal eye circuit shown in relation to the outline of neuropil, reconstructed by ssTEM. The position of synapses in the visual circuit reveals the primary and secondary optic neuropils. The schematized ciliary band cells are also shown. (**C**) ssTEM reconstruction of photoreceptors, glia cells and synapses. (**D**) ssTEM reconstruction of photoreceptors and primary interneurons (IN¹). (**E**) ssTEM reconstrucion showing the trans-optic-neuropil interneurons (INᵗᵒⁿ) connecting the two optic neuropils. (**F**) Confocal microscopic image of a 3-day-old larva stained with phalloidin to label the musculature (red) and with DAPI to label nuclei (cyan). (**G**) ssTEM reconstruction of muscles and motorneurons (MN). (**H**) ssTEM reconstruction of the complete cell complement of the minimal visual circuit. Neurons are colored by type. Pigment cups are shown in brown. All images show anterior views. PRC, photoreceptor; IN, interneuron; MN, motorneuron; eyeᵃˡ, anterior-left eye; eyeᵃʳ, anterior-right eye; eyeᵖˡ, posterior-left eye; eyeᵖʳ, posterior-right eye; ON, optic neuropil; cPRC, ciliary photoreceptor; DCC, dorsal branch of the circumesophageal connectives; VCC, ventral branch of the circumesophageal connectives; NSP, neurosecretory plexus; DLM, dorsal longitudinal muscle; VLM, ventral longitudinal muscle. The coloring of cell types is consistent throughout the paper (PRC, blue; IN¹, yellow; INⁱⁿᵗ, lilac; INᵗᵒⁿ, magenta; INᵈᶜ, light brown; INᵛᶜ, cian; INˢⁿ, green; MN, red). Scale bars, 30 µm. The Blender atlas with the volume rendering of all cells and synapses is available in *Randel et al. (2014)*.

The following figure supplements are available for figure 2:

**Figure supplement 1**. Muscles and ciliary bands in 3-day old larvae.

using whole-mount RNA in situ hybridizations and image registration, allowing the molecular profiling of cell types (*Tomer et al., 2010*; *Asadulina et al., 2012*). Although expression patterns for several of these genes have already been reported in *Platynereis* (*Denes et al., 2007*; *Jékely et al., 2008*; *Tomer et al., 2010*), we acquired new whole-body expression data to analyze patterns in the 3-day-old brain in more detail. We found that *vesicular glutamate transporter* (*VgluT*), a marker of glutamatergic neurons, colocalized with *r-opsin1*, a marker of eye photoreceptors (*Arendt et al., 2002*; *Randel et al., 2013*; *Figure 3—figure supplement 2*, *Figure 3—figure supplement 3*), indicating that the photoreceptors are glutamatergic. In contrast, *histidine decarboxylase* (*hdc*), an enzyme catalyzing the synthesis of histamine, the transmitter in arthropod rhabdomeric photoreceptors (*Stuart, 1999*), does not localize to the eyes, as shown by image registration and double RNA in situ hybridization (*Figure 3—figure supplement 3*).

In the cell body regions of the interneurons we found expression of several monoaminergic markers in single cells or small cell clusters. These include *tryptophan hydroxylase* (*TrpH*), *tyrosine hydoxylase* (*TyrH*), *hdc*, *dopa-beta-hydroxylase* (*dbh*), markers for serotonergic, dopaminergic, histaminergic, and noradrenergic neurons, respectively (*Figure 3B*, *Figure 3—figure supplement 2*, *Figure 3—figure supplement 3*). This indicates that the interneurons of the eye circuit use several distinct monoamine neurotransmitters.

*Choline acetyltransferase* (*ChAT*) and *vesicular acetylcholine transporter* (*VAChT*), two markers of cholinergic neurons, are localized in the cell body region of the ventral motorneurons (*Figure 3C*,

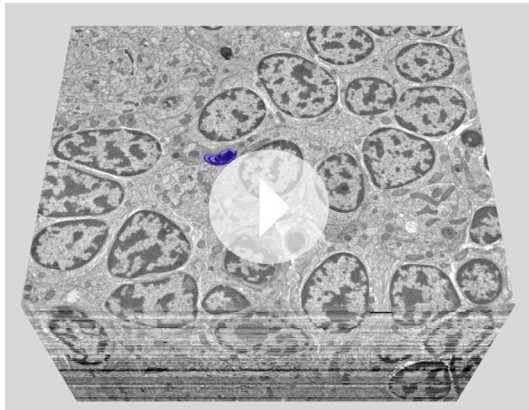

**Video 4**. Volume rendering of the left primary optic neuropil. The reconstructed volume shows parts of two glia cells (pink and light blue), one IN[1] axon (yellow) and the photoreceptor axons (different hues of blue) forming synapses on the IN[1] axon. The position of the individual synapses to the IN[1] axon are indicated in different colors for the different photoreceptors.

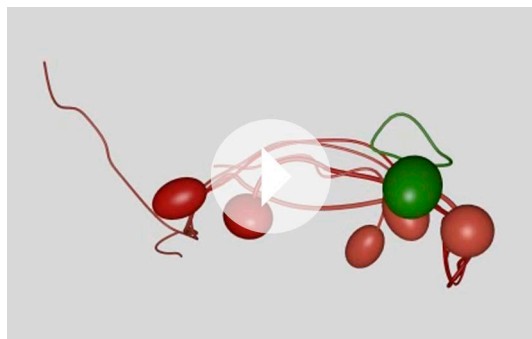

**Video 5**. Volume reconstruction of a Schnörkel interneuron and motorneurons. 3D reconstruction of one Schnörkel interneuron and its postsynaptic motorneuron targets. One Schnörkel interneuron is shown with its five postsynaptic motorneurons. The synapses are indicated. The Schnörkel interneuron connects to both ipsilateral and contralateral motorneurons either at a cell body proximal or a cell body distal position along the motorneuron axon.

*Figure 3—figure supplement 2*). This is consistent with earlier observations showing that the longitudinal trunk muscles, targets of the ventral motorneurons, are influenced by acetylcholine (*Denes et al., 2007*). A marker for GABAergic cells, *glutamate decarboxlase* (*gad*), partly colocalizes with the ventral cholinergic domain, in the area of the motorneuron cell bodies (*Figure 3C*, *Figure 3—figure supplement 2*, *Figure 3—figure supplement 3*). This suggests that some motorneurons may have mixed GABAergic and cholinergic identity. The difference in mean vesicle size at neuromuscular and neurociliary synapses in motorneurons (*Figure 3—figure supplement 1*) also supports a mixed neurotransmitter identity. Further cellular resolution mapping will be needed to confirm this and to establish the neurotransmitter identities of individual interneurons and motorneurons.

## Network analysis of the eye connectome

For a detailed analysis of synaptic contacts in the eye circuit, we represented the connectome as a directed graph and as an all-against-all synaptic connectivity matrix (*Figure 4A*, *Figure 4—figure supplement 1*, *Figure 4—source data 1*). In the graph, the nodes correspond to neurons and the directed edges to synaptic contacts, weighted by synapse number. We displayed the full cell complement (79 nodes; *Figure 4A*) and a trimmed graph (61 nodes) where all cells with <3 pre- and post-synaptic sites were omitted and only edges of three or more synapses were shown (*Figure 4—figure supplement 2*). Manual or force-field-based clustering was used for graph layout. We also constructed a merged graph where the same neuron types from the left and right body sides were grouped. In this graph the weight of connections is represented as the maximum number of synapses between any individual cell pair of the distinct groups (*Figure 4B*, *Figure 4—source data 2*).

To analyze information flow we calculated the maximum distance of all nodes to any other node within the directed graph (eccentricity; *Figure 4A*). The eye circuit is a feed-forward circuit where information flows from the eyes to the muscles and ciliary bands. We also calculated centrality measures to identify strongly connected nodes in the network (*Figure 4—figure supplement 2*). With three different centrality measures (eigenvector centrality, weighted-degree centrality and authority) the four primary interneurons (IN[1]) ranked among the top five nodes (*Randel et al., 2014*). These four cells receive several synapses from the photoreceptors, with the highest number of each IN[1] received from the crosswise eyes (*Figure 4B*, *Figure 4—figure supplement 2*). The four IN[1] cells also have strong mutual connections (see below).

A search for highly interconnected nodes or communities (*Blondel et al., 2008*) subdivided the eye network into four modules (*Figure 4—figure supplement 3*). The four IN[1] cells belong to two different

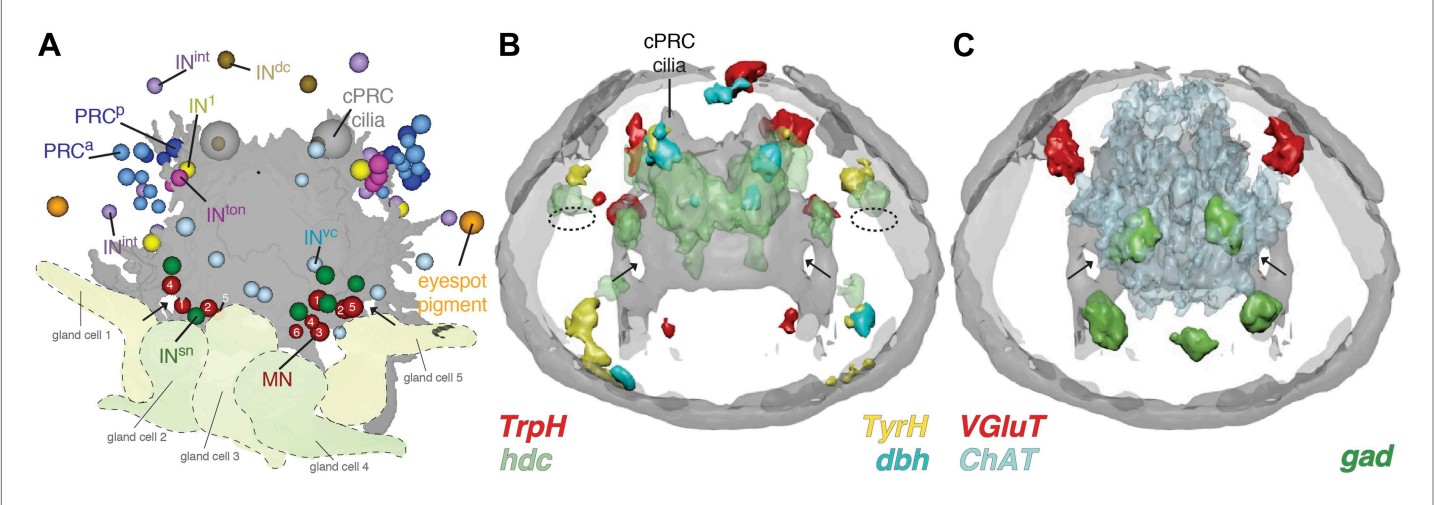

**Figure 3**. Neurotransmitters of the eye circuit. (**A**) Cell body positions of eye circuit neurons relative to the larval axonal scaffold and five large gland cells. Motorneurons are numbered according to the cell identifiers. (**B** and **C**) Surface representation of the average expression domains of neurotransmitter marker genes relative to the larval axonal scaffold. The following genes are shown: (**B**) histaminergic marker *histidine decarboxylase* (*hdc*; green), serotonergic marker *tryptophan hydroxylase* (*TrpH*; red), dopaminergic marker *tyrosine hydroxylase* (*TyrH*; yellow), adrenergic marker *dopamine beta hydroxylase* (*dbh*; cyan), (**C**) glutamatergic marker *vesicular glutamate transporter* (*VGluT*; red), cholinergic marker *choline acetyltransferase* (*ChAT*; grey), GABAergic marker *glutamate decarboxylase* (*gad*; green). The axonal scaffold, based on average acetylated-tubulin signal, is shown in grey. PRC, photoreceptor; cPRC, ciliary photoreceptor; IN, interneuron; MN, motorneuron. In (**B**) dashed ovals mark the position of the eyespots. Black arrows show the ring formed by the circumesophageal connectives.

The following figure supplements are available for figure 3:

**Figure supplement 1**. Synaptic vesicle diameter for different synapse types.

**Figure supplement 2**. Expression of neurotransmitter markers in the head of *Platynereis* larva.

**Figure supplement 3**. Neurotransmitter marker gene expression profiling.

modules in the eye circuit (*Figure 4—figure supplement 3*), which together contain all neurons with projections intrinsic to the primary optic neuropil. Both modules consist of a crosswise pair of eyes and the associated IN¹ cells and trans-optic-neuropil interneurons (INᵗᵒⁿ). The anatomy of the circuitry in the primary optic neuropil thus displays point symmetry rather than bilateral symmetry (*Figure 4—figure supplement 3*).

The other two modules, the motor modules, contain all neurons intrinsic to the secondary optic neuropil, including Schnörkel interneurons and motorneurons and associated effector organs. A bilateral pair of INᵗᵒⁿ cells links the modules of the primary and secondary optic neuropils, reflecting the anatomy where INᵗᵒⁿ cells receive synapses in the primary and project to the secondary optic neuropil (*Figure 4—figure supplement 3*). One of the motor modules provides innervation to the prototroch and metatroch ciliary band cells. The other module contains motorneurons that connect to three out of four bundles of longitudinal muscles (we could not find neuromuscular synapses on the left dorsal longitudinal muscle) (*Figure 4A,B*, *Figure 4—figure supplement 2*).

The connectome reveals that all four eyes can provide motor input to both body sides. The neuronal pathways from the eyes cross the midline multiple times throughout the circuit. The strongest synaptic connections are formed between photoreceptor—IN¹—INᵗᵒⁿ—Schnörkel interneuron, always to the contralateral side in a feed-forward circuit (*Figure 4B*). However, most Schnörkel interneurons form several synapses on both ipsilateral and contralateral motorneurons in the secondary optic neuropil, representing a bilateral divergence of the connections. Motorneuron axons receive cell-body-proximal synapses from the ipsilateral Schnörkel interneurons, cross the midline, and receive cell-body-distal synapses on the other side of the body from the contralateral Schnörkel interneurons (*Figure 4—figure supplement 6G,H*; *Video 5*).

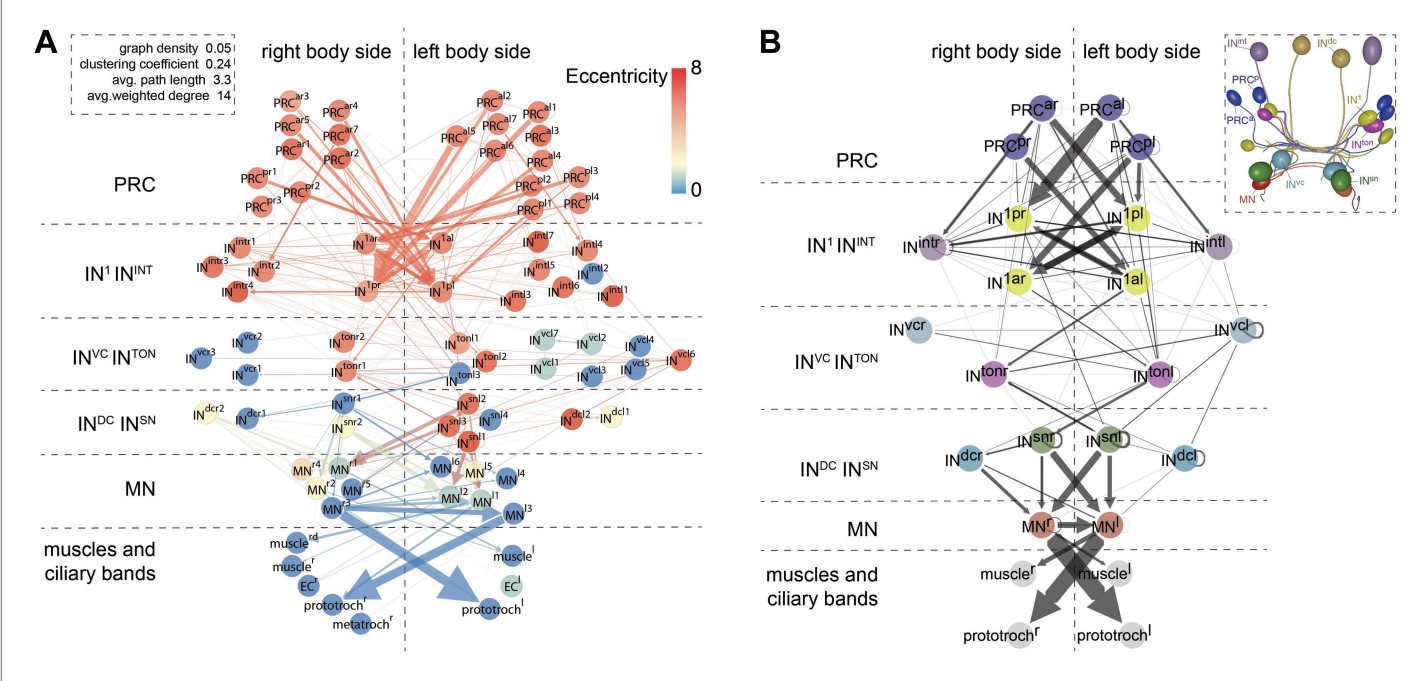

**Figure 4**. Network analysis of the visual eye circuitry. (**A**) Full connectomic graph of the visual eye circuit including 71 neurons and 8 effectors (muscles, ciliary band cells and epithelial cells). The edges are directed from presynaptic cell pointing to postsynaptic cells. Edges are weighted by the number of synapses. Inset shows selected network parameters. (**B**) Merged graph representation of the visual circuit. Nodes correspond to neuron classes, edges are weighted by the maximum number of synapses between two neuron types of each class. Nodes are colored following the color scheme used to label cell types. Inset shows the anatomical position of the cell types. PRC$^{al}$, anterior-left photoreceptors; PRC$^{ar}$, anterior-right photoreceptors; PRC$^{pl}$, posterior-left photoreceptors; PRC$^{pr}$, posterior-right photoreceptors; IN, interneuron; MN, motorneuron. Matrix files of the complete and the merged networks are available in *Figure 4—source datas 1 and 2*.

The following source data and figure supplements are available for figure 4:

**Source data 1**.

**Source data 2**.

**Figure supplement 1**. Synaptic connectivity matrix of the *Platynereis* larval visual circuit.

**Figure supplement 2**. Connectivity graphs of the *Platynereis* larval visual circuit.

**Figure supplement 3**. Modules in the eye connectivity graph.

**Figure supplement 4**. Connectivity matrix of the left and right body sides.

**Figure supplement 5**. Stereotypy of synapse distribution on IN$^1$ cells.

**Figure supplement 6**. Stereotypy of synapse distribution on IN$^{ton}$, IN$^{sn}$ and MN cells.

## Stereotypy of synaptic connections

In order to estimate how stereotypic synaptic connectivity is within the circuit we compared the connectivity matrices of neurons with cell bodies on the left or right side of the body. We used the merged graph (*Figure 4B*) since not all neurons had a contralateral pair (e.g., we identified three IN$^{ton}$ cells on the left and two on the right body side). We found a strong correlation between the left and right connectivity matrices (Spearman r = 0.67, p<0.0001; *Figure 4—figure supplement 4*).

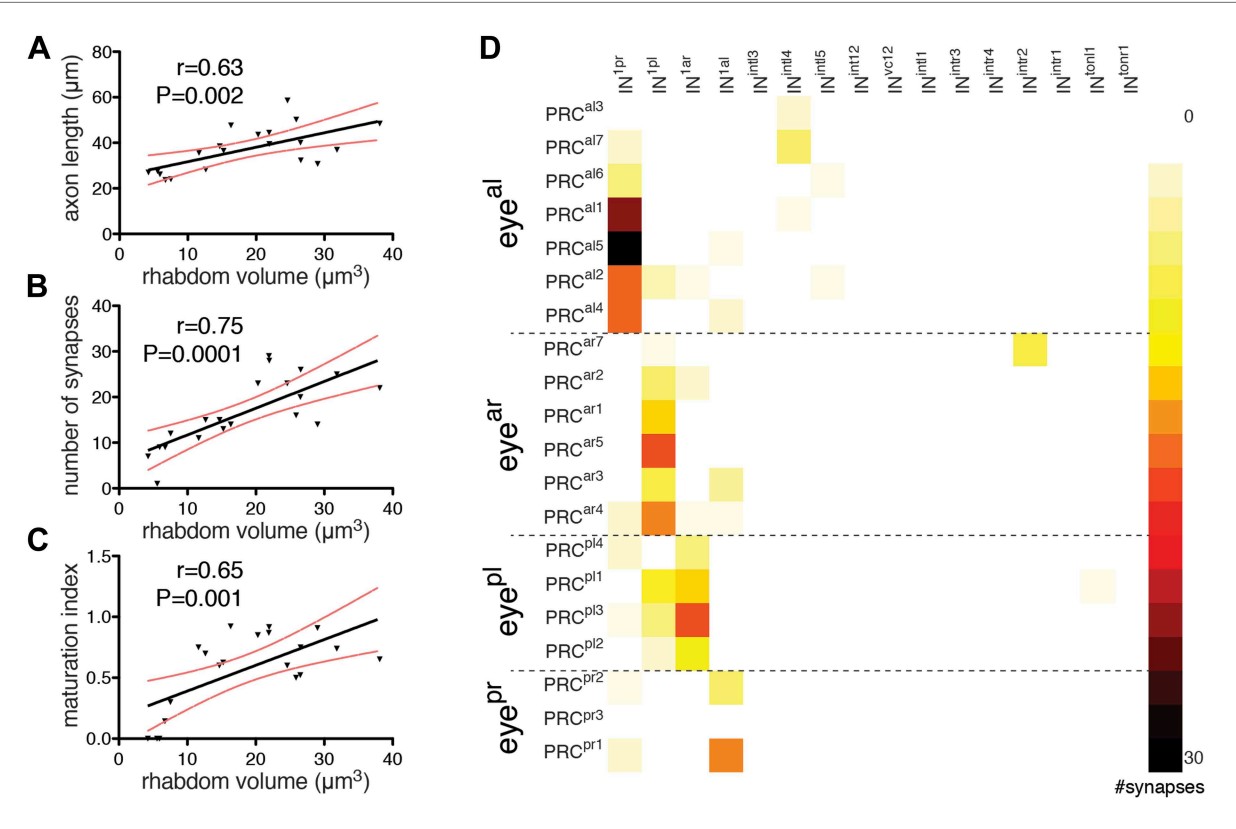

**Figure 5**. Maturation of photoreceptor connections. (**A**) Relationship of rhabdom volume to photoreceptor axon length. (**B**) Relationship of rhabdom volume to photoreceptor synapse number. (**C**) Relationship of photoreceptor connectivity-maturation index to rhabdom volume. In **A–C** the black line shows linear regression with 95% confidence interval (red dashed lines). Pearson r and p-value are shown. (**D**) Connectivity matrix of the photoreceptors. The matrix is ordered from top to bottom by eye and then for each eye by photoreceptor rhabdom size increasing from top to bottom. Eye$^{al}$, anterior-left eye; eye$^{ar}$, anterior-right eye; eye$^{pl}$, posterior-left eye; eye$^{pr}$, posterior-right eye.

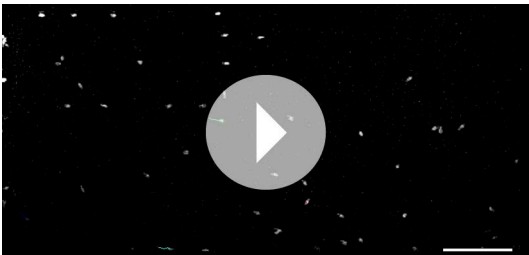

**Video 6**. Mixed positive and negative phototaxis in 3-day-old *Platynereis* larvae. Larvae were stimulated with alternating directional white light from the left or the right side of the phototaxis cuvette (shown by white bars on the side). Larvae display mixed phototaxis, some negatively phototactic larvae are tracked. Scale bar, 2 mm. Time increment: 0.07 s.

We also analyzed the stereotypy of the spatial arrangement of pre-and postsynaptic sites among neurons of the same type. The four IN[1] cells always receive photoreceptor input in a cluster at the most distal segment of their axon (*Figure 4—figure supplement 5*). Synapses onto the IN[1] cells from the contralateral photoreceptors are intermingled with presynaptic sites of IN[1] cells to IN[ton] cells (*Figure 4—figure supplement 6*) and are not spatially segregated within the cluster. In contrast, IN[1] cells receive axo-axonal synaptic input from the crosswise IN[1] cell in a cluster positioned more proximal to the cell body (*Figure 4—figure supplement 5*). The spatial arrangement of synapses is also similar between the left and right sides for other neuron types (*Figure 4—figure supplement 6*). These observations show that both the number and the spatial arrangement of synapses are stereotypic between the left and right body sides. The reconstruction of further individuals will be needed to assess individual-to-individual variation.

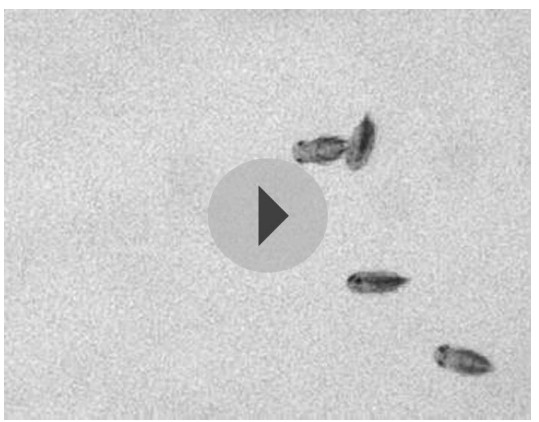

**Video 7**. A single phototactic turn in 4-day-old larvae. The larvae were exposed to a 180-degree change in the direction of the white stimulus light, eliciting a phototactic turn. The stimulus direction is indicated by the white bars at the side of the video. During the turn the larvae are bending due to the contraction of the longitudinal muscles on one body side. Time increment: 0.066 s.

## Maturation of photoreceptor connectivity

During development of the *Platynereis* larva, photoreceptors are continuously added to the eyes at the periphery of the pigment cup (*Rhode, 1992*). Individual photoreceptors in an eye are thus at different stages of differentiation within the same individual. We noted that in our reconstructed specimen the photoreceptor rhabdoms have different sizes, and that rhabdom volumes positively correlate with axon length and synapse number (*Figure 5A,B*). Using rhabdom volume as a proxy to photoreceptor differentiation we then analyzed how photoreceptor connections change during neuron maturation (*Figure 5D*).

Photoreceptors with short axons connect weakly if at all to the primary interneurons (IN[1]), while photoreceptors with longer axons form more synapses on IN[1] cells. Photoreceptors with the shortest axons have connections to other interneurons (e.g., IN[intl4], IN[intr2]) that are not observed in photoreceptors with longer axons (*Figure 5D*). We defined a photoreceptor connectivity-maturation index (number of crosswise IN[1] connections per all connections) and found that it correlates with rhabdom volume (*Figure 5C*). These data represent a snapshot during development in one individual, but suggest that during photoreceptor development connections to the IN[1] cells get stronger and initial contacts to other interneurons are eliminated.

The IN[ton] cells also show variation in axon length and connectivity, possibly also reflecting developmental progression. The small number of these interneurons precluded similar analyses.

## The eyes mediate visual phototaxis

The innervation of the longitudinal trunk muscles and ciliary bands by the eye circuit motorneurons suggested that the eyes could regulate tail bending and ciliary beating during larval swimming. To understand the function of the eye circuit we next analyzed larval photobehavior in detail, in conjunction with eye ablation experiments.

We first analyzed swimming trajectories of freely behaving 3- and 4-day-old larvae exposed to light stimuli of alternating directionality. 3- and 4-day-old larvae swim using their cilia while rotating around their anterior–posterior axis. When we used alternating illumination from the two opposite sides of an assay cuvette we observed directional swimming (phototaxis) of the larvae. Larvae showed mixed behavior, with some swimming towards and others away from the light source (*Video 6*). Such sign-switch in directional swimming responses is common for marine larvae and can be influenced by various environmental stimuli (*Thorson, 1964*; *Young and Chia, 1982*; *Marsden, 1990*). During reorientations we could observe prominent body bending in 4-day-old larvae and weaker bending in 3-day-old larvae, suggesting that larvae turn by contracting longitudinal muscles while swimming with cilia (*Video 7* and data not shown). In agreement with the cholinergic identity of the motorneurons, treatment with mecamylamine, an acetylcholine receptor antagonist, blocked negative phototaxis. Mecamylamine treatment increased swimming speed, probably via influencing cilia (*Jékely et al., 2008*). The effects could be reversed by washout (*Figure 6—figure supplement 1*; *Randel et al., 2014*).

Next we illuminated the cuvette constantly from both sides but with different left-right light intensities and measured the efficiency of phototaxis for a population of larvae using a phototaxis index. We found that phototactic efficiency increased with increasing contrast between the light intensity at each side of the cuvette, but was independent of total light intensity (*Figure 6—figure supplement 2*).

To test the contribution of the eyes and the two eyespots (independent ventral structures that develop in 1-day-old larvae, *Figure 1B*) to directional swimming, we laser-ablated the eyes or eyespots and subsequently exposed the larvae to directional illumination. Ablation of the two eyespots required for non-visual phototaxis in 1-day-old larvae (*Jékely et al., 2008*) did not abolish phototaxis responses

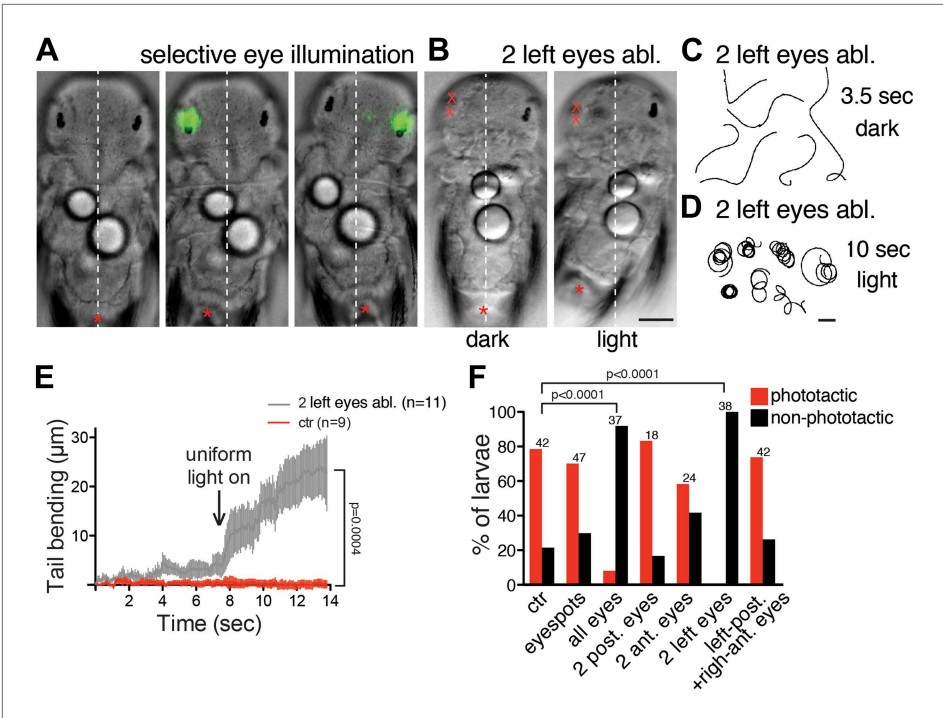

**Figure 6**. Eyes mediate body bending during visual phototaxis. (**A**) Selective eye illumination triggers body bending in an immobilized larva. (**B**) A larva with both left eyes ablated displays body bending upon uniform illumination with white light (light) but not with a red filter (dark). An asterisk marks the tip of the tail. (**C** and **D**) Trajectories of larvae with both left eyes ablated in the dark (red filter) (**C**) and upon uniform illumination with white light (**D**). (**E**) Tail bending upon uniform white light illumination of non-ablated control larvae and larvae lacking the two left eyes. Data are shown as mean ± SEM, n >9 for both condition. p-value of a *t* test calculated for the last time point is indicated. (**F**) Percentage of phototactic (red) and non-phototactic (black) larvae among non-ablated control and various eye-ablated larvae. p-values of a chi-square test are indicated relative to non-ablated controls. Only the 'all eyes' and 'two left eyes' ablated conditions are significantly different from non-ablated control. Number of larvae tested is shown above the columns for each condition. Scale bars, 40 μm (**A** and **B**), 1 mm (**C** and **D**).

The following figure supplements are available for figure 6:

**Figure supplement 1**. Inhibition of phototaxis by a cholinergic antagonist.

**Figure supplement 2**. Efficiency of phototaxis depends on contrast, not absolute light levels.

in 3-day-old larvae. In contrast, ablation of all four eyes led to the loss of directional swimming (*Figure 6F*), demonstrating that only the eyes mediate this behavior.

To directly investigate if the eyes are able to mediate body bending we performed selective eye illumination experiments on larvae held between a slide and a coverslip. Unilateral illumination of the eyes triggered reproducible bending of the body in 3-day-old larvae (*Figure 6A*; *Video 8*). When we used varying stimulus-light intensities for selective eye illumination we obtained graded bending responses with saturation kinetics (*Figure 7D*).

Unilateral laser ablation of both eyes (*Video 9*) followed by uniform illumination also triggered strong and persistent body bending (*Figure 6B,E*; *Video 10*). Interestingly, in both experiments, the larvae either bent on or opposite to the illuminated side, with some larvae switching the response during repeated stimulation (*Video 8* and data not shown). These results indicate that it is either the contra- or the ipsilateral motor pathway that dominates. Given that the Schnörkel interneurons synapse on both contra- and ipsilateral motorneurons (*Figure 4—figure supplement 6*), phototactic sign switching may take place at the level of the motorneurons.

To dissect the interplay of the four eyes during visual phototaxis, we ablated them in different combinations and tested phototactic ability. Larvae with unilateral ablation of both eyes were not

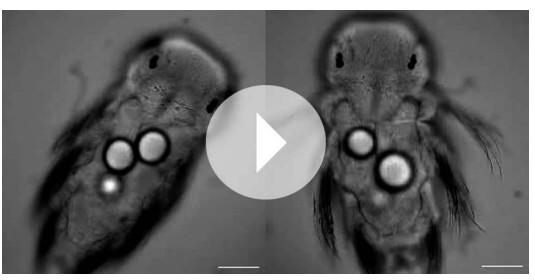

**Video 8**. Selective illumination of eyes triggers body bending. Illumination of the eyes on one body side in the area shown by the green signal triggers body bending on the opposite body side, corresponding to negative phototaxis (left), or the same body side, corresponding to positive phototaxis (right). Scale bar, 50 µm. Time increment: 0.43 s.

phototactic and swam persistently in circles when illuminated (*Figure 6C,D,F*). The two anterior and two posterior eyes connect to distinct, though partly overlapping, circuits. To test if input to both the anterior and posterior eyes is necessary for directional swimming, we ablated either the anterior or the posterior eye pair. We also performed crosswise eye ablations. All of these ablated larvae showed phototactic responses similar to non-ablated controls (*Figure 6F*), hence the presence of at least one eye on both body sides is necessary and sufficient for directional turns.

To investigate if both dorsal and ventral longitudinal muscles are involved in body bending, we performed calcium-imaging experiments using ubiquitously expressed GCaMP6 (*Chen et al., 2013*). In immobilized larvae we repeatedly stimulated the eyes with 488 nm light on one body side. In agreement with the circuit diagram, upon selective eye stimulation we could observe corresponding calcium signals in both the ventral and dorsal longitudinal muscles on the opposing body side (*Videos 11 and 12*).

Overall, these experiments demonstrate that the eyes mediate visual turns by comparing simultaneous light inputs at each side of the body. This, together with the shading provided by the pigment cups (*Video 2*), allows the detection of the spatial distribution of light, without body movement. The visual contrast then leads to the graded contraction of the longitudinal muscles during turns. The modulation of ciliary beating, not investigated here, may also contribute to phototactic turns, as is known for younger larvae (*Jékely et al., 2008*). This sensorimotor structure is similar to visual phototaxis described in the lamprey (*Ullén et al., 1997*), but is very different from the non-visual phototactic responses in *Drosophila* larvae (*Kane et al., 2013*). Visual phototaxis is also fundamentally different from the helical phototaxis mediated by the eyespots during early *Platynereis* larval stages (1–2 days) (*Jékely et al., 2008*).

## A contrast-enhancing circuit motif

When we analyzed mutual synaptic connections in the eye circuit we found that by far the strongest mutual connections are formed between the crosswise pairs of IN[1] cells (up to 18 synapses one way; *Figure 7A,B*). These mutual, and thus likely inhibitory, connections are formed at the cell-body proximal segment of the IN[1] cells' axons, spatially segregated from the more distal sites of photoreceptor input (*Figure 4—figure supplement 5*). This reciprocal wiring may provide mutual inhibition and reinforce small differences in activation between the crosswise eyes, potentially representing a network motif for enhancing contrast-detection. Modularity analysis also revealed that the sub-networks of the crosswise eyes are more strongly connected than those of the pairwise eyes (*Figure 4—figure supplement 3*). This suggests that two reciprocally connected crosswise eyes are more efficient in visual information processing than a bilateral pair.

To test this, we performed various eye ablations and then subjected the larvae to selective eye illumination. We ablated a bilateral pair of eyes, thus eliminating inputs to one side of both crosswise mutual IN[1] motifs. We expect that these larvae would have a reduced bending response at the same level of contrast. We also performed crosswise eye ablations, leaving input to one of the crosswise IN[1] motifs intact. We then exposed larvae to a 488 nm background illumination and measured their tail bending following unilateral eye illumination with 488 nm stimulus light of varying intensity. Larvae with pairwise ablation of the anterior eyes showed significantly reduced bending relative to non-ablated controls with the average maximum tail displacement reduced to approximately half (*Figure 7C,D*). In contrast, larvae with crosswise eye ablation showed stronger bending that was not significantly different from non-ablated controls. Although both crosswise and pairwise ablated larvae have only two eyes and are phototactic (*Figure 6F*), these results show that pairwise ablated larvae have a markedly reduced motor response at the same level of contrast. We conclude that the strong mutual contacts between the crosswise IN[1] cells represent a circuit motif that enhances turning responses during phototaxis.

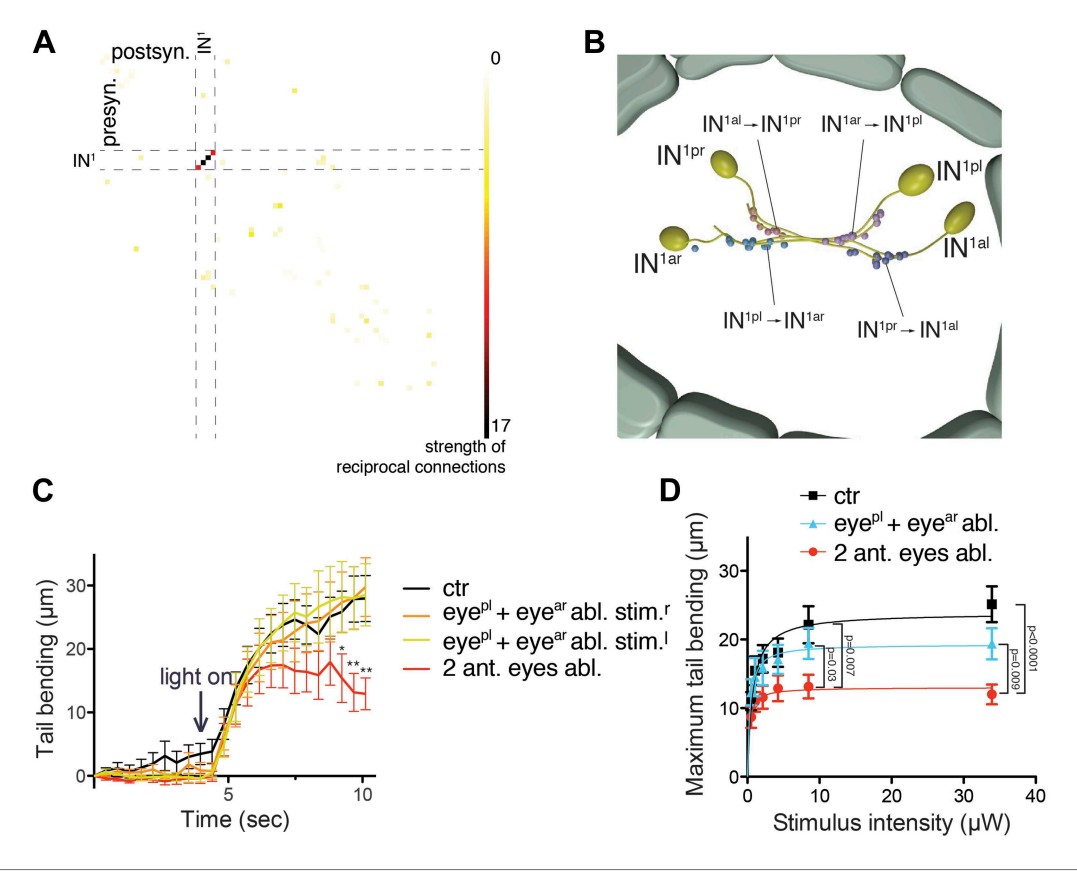

**Figure 7**. An interneuron motif for enhanced contrast detection. (**A**) Strength of reciprocal connections between all neuron pairs in the complete visual circuit. The strength of reciprocal connections was defined as the geometric mean of the number of reciprocal synapses between each neuron pair. The single neuron identifiers are not shown for simplicity. (**B**) The strongest reciprocal motif in the eye circuit is between the crosswise IN[1] pairs. The position and polarity of the synapses are indicated. (**C**) Quantification of tail bending in different eye ablated larvae under selective eye illumination with a 488-nm laser. Data are shown as mean ± SEM, n >17 for each condition. Larvae lacking two anterior eyes were compared to non-ablated control larvae at each time point (*p value<0.05, **p value<0.01, unpaired *t* test). (**D**) Signal-response curve of maximum tail bending upon selective eye illumination in immobilized larvae using 488-nm stimuli of different intensities. The data are fitted with a saturation-binding curve. Data are shown as mean ± SEM, n >24 larvae for each condition (independent from **C**). p-values of unpaired *t* tests comparing larvae lacking two anterior eyes to the other conditions are shown. Source bending data from (**C** and **D**) are shown in *Figure 7—source data 1* and *Figure 7—source data 2*.

The following source data are available for figure 7:

**Source data 1**.

**Source data 2**.

## Discussion

Here we described the neural connectome of the visual system in the *Platynereis* larva. The four eyes mediate visual phototaxis, during which larvae are able to detect and contrast spatial differences in the light field. Although it is not yet possible to directly link all neurons and connections in the *Platynereis* larva to phototactic behavior, the combined analysis of connectivity and behavior yielded fundamental insights into a visually guided behavior.

We concluded that spatial vision is not due to the presence of more (3–7 in our larva) photoreceptors in the eyes, since all photoreceptors from one eye synapse on the same primary interneuron. Instead, spatial vision relies on at least two eyes, pointing in different directions, and the underlying neural circuit

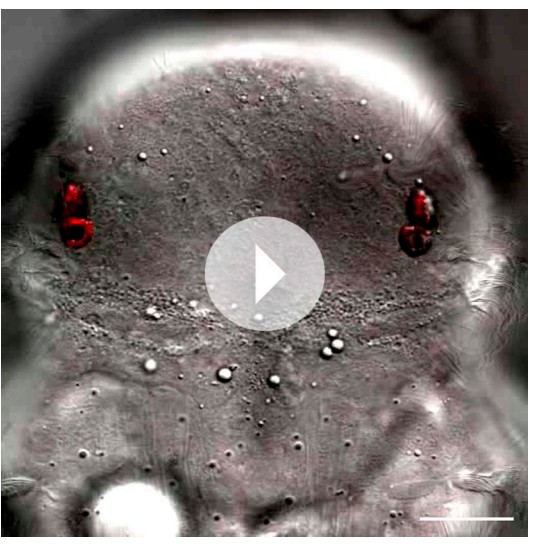

**Video 9**. Laser ablation of the eyes. The position of the four eyes and the two eyespots is shown by changing the imaging focus. The right eyes are ablated. The eye pigment is imaged using reflection imaging (red), and is overlaid on the DIC channel. Scale bar, 25 µm. Time increment: 1.1 s.

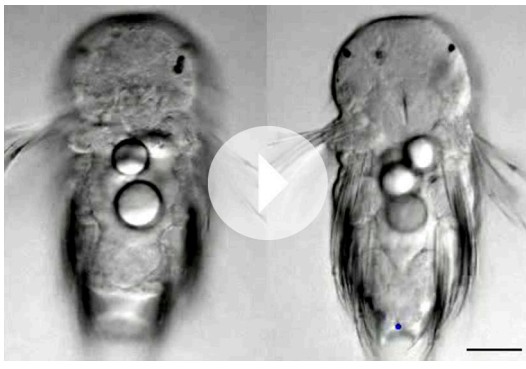

**Video 10**. Uniform illumination triggers body bending following unilateral eye ablation. Both left eyes were ablated. The larva is imaged with DIC illumination with a 750 nm long-pass filter. The filter is removed to provide uniform white illumination from the microscope lamp at frame 98. The pigment of the right eye is visible. Scale bar, 50 µm. Time increment: 0.1 s.

that contrasts simultaneous light inputs at the left and right eyes. Furthermore, connectomics and modularity analysis revealed that the four eyes connect to the downstream visual circuitry in a crosswise manner, showing point symmetry. The crosswise-eye-modules show mutual connections at the level of the primary interneurons. In agreement with this, ablation experiments demonstrated that a bilateral pair of eyes elicits a smaller motor response than a crosswise pair, at the same level of light-intensity contrast. The contrast enhancement takes place in the primary optic neuropil circuitry, and the $IN^{ton}$ may relay a contrasted, one-sided signal to the secondary optic neuropil.

In the secondary optic neuropil this one-sided input could lead to the biased activation of the left or right motorneurons. Our circuit reconstructions partly explain the bimodality of the behavior (either positive or negative phototaxis). Such switching behavior requires neuronal connections between the eyes on one body side and the muscles on both body sides. We found that indeed the circuit diverges bilaterally at the level of the Schnörkel interneurons ($IN^{sn}$), when these cells synapse to motorneurons of both the left and right body side. The Schnörkel interneuron synapses form at axon segments either proximal or distal to the motorneuron cell bodies (***Figure 4—figure supplement 6***). The spatial organization of synapses may provide an initial bias to the system, favoring bends on one body side. Further modulatory input may influence this bias, inducing a sign switch. There are several sensory neurons, not described here, that feed into the minimal eye circuit at the middle segment of the motorneuron axons (NR, LABC, and GJ, unpublished). Further work will be needed to characterize the possible roles of these neurons in sign switching.

The sensory-motor strategy of visual phototaxis is similar to that found in the lamprey (***Ullén et al., 1997***; ***Figure 8***). In this vertebrate, unilateral illumination leads to a lateral turn away from the light during negative phototaxis. However, under some circumstances lampreys display positive phototaxis. Lesion experiments demonstrated the involvement of the pretectum and reticulospinal neurons in phototaxis, the latter forming the descending control system. Surgically severing connections between the pretectum and the ipsilateral reticulospinal neurons (***Figure 8E***) leads to a sign switch in the phototactic response when the ipsilateral side is illuminated. Transection of the ventral rhombencephalic commissure (***Figure 8E***) in turn reduces the turning angle during negative phototaxis. Thus both the *Platynereis* and the lamprey phototactic circuits are characterized by extensive midline crossing at several levels, bilateral divergence to allow context-dependent sign switching, and bilateral, probably inhibitory, interactions to enhance turning magnitude (***Figure 8D,E***).

At the cellular level, the *Platynereis* eye circuit also shows similarity in its multi-layered arrangement to the visual circuits of insects and vertebrates (***Figure 8A–C***; ***Sanes and Zipursky, 2010***). In the visual system of *Drosophila* photoreceptors project to the optic lobe that is organized into distinct ganglia

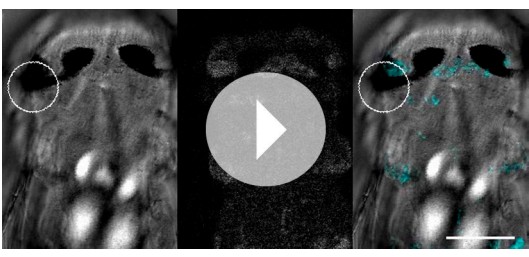

**Video 11**. Calcium-imaging in ventral longitudinal muscles. Calcium-imaging with GCaMP6 in the ventral longitudinal muscle during selective illumination of the right eyes. The larva is ventrally oriented, the eyes are out of focus. The circular ROI shows the illuminated area. The duration of the illuminations is visible in the DIC channel (brighter overall signal). Scale bar, 50 μm. Time increment: 0.57 s.

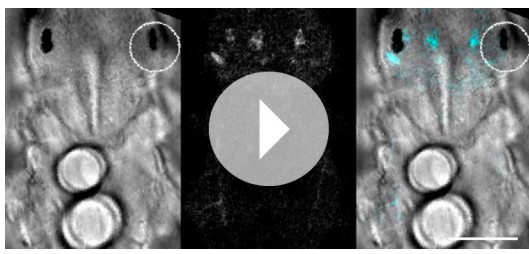

**Video 12**. Calcium-imaging in dorsal longitudinal muscles. Calcium-imaging with GCaMP6 in the dorsal longitudinal muscle during selective illumination of the right eyes. The larva is dorsally oriented, the eyes are visible. The circular ROI shows the illuminated area. The duration of the illuminations is visible in the DIC channel (brighter overall signal). Scale bar, 50 μm. Time increment: 0.57 s.

(the lamina, medulla and lobula complex) (*Erclik et al., 2009*). Some neurons are intrinsic to one ganglion, others connect two adjacent ganglia (e.g., transmedullary neurons). Similarly, in the *Platynereis* circuit the IN[1] and IN[int] cells are intrinsic to the primary optic neuropil, and IN[ton] cells link the primary and secondary neuropils. With the exception of the photoreceptors (*Arendt et al., 2002*), the evolutionary relationships of the cell types ('cell-type homology') (*Arendt, 2008*; *Erclik et al., 2009*) in the vertebrate, insect, and annelid visual systems are unclear and more comparative work will be needed to assess the evolutionary significance of these similarities.

Additionally, our work provides a more general insight about the evolution of higher-resolution image-forming eyes. The current model of eye evolution defines four steps (*Nilsson, 2009*), from non-directional photoreception through directional scanning photoreception (spiral phototaxis) (*Jékely et al., 2008*; *Jékely, 2009*) and low-resolution spatial vision to high-resolution spatial vision. We now extend this scheme with the concept of the two-pixel visual phototactic eye, likely predating the evolution of low-resolution spatial vision.

Simple eyes, similar to the eyes of *Platynereis* larvae, are widespread in the planktonic larval stages of several bilaterians and may often function in visual phototaxis (*Buchanan, 1986*; *Blumer, 1994*; *Lacalli, 2004*). More complex image-forming eyes may have repeatedly evolved from such phototactic eyes. Larval eyes sometimes directly develop into the image-forming eyes of the adults (*Cazaux, 1985*; *Blumer, 1994*). In *Platynereis*, and several other annelids and mollusks, the larval eyes develop into the adult's eyes, which harbor hundreds or thousands of photoreceptors (*Rhode, 1992*) and mediate low-resolution image-forming vision. In *Platynereis*, we identified visual phototaxis as the first function during the development of the eyes. As in development, also during evolution the first images seen by animals may have consisted of a dark field at the bottom and a bright field at the top of the ocean.

## Materials and methods

### Animal culture

*Platynereis* larvae were obtained from an in-house breeding culture following an established protocol (*Hauenschild and Fischer, 1969*). Larvae were kept at 18°C for development. Behavioral experiments were performed at room temperature.

### ssTEM

72 hr post fertilization *Platynereis* larvae, reared at 18°C, were fixed using a high-pressure freezer (HPM 010; BAL-TEC, Balzers, Liechtenstein) and transferred to liquid nitrogen. Frozen samples were cryosubstituted with 2% osmium tetroxide in acetone and 0.5% uranyl acetate in a cryosubstitution unit (EM AFS-2; Leica Microsystems GmbH, Wetzlar, Germany) over a regime of gradually rising temperatures. Samples were embedded in Epon. 40–50-nm serial sections were cut starting from the anterior end of a larva (*Platynereis* HT-9-3) on a Reichert Jung Ultracut E microtome. The sections were collected on single-slotted copper grids (NOTSCH-NUM 2 × 1 mm, Science Service, Munich, Germany)

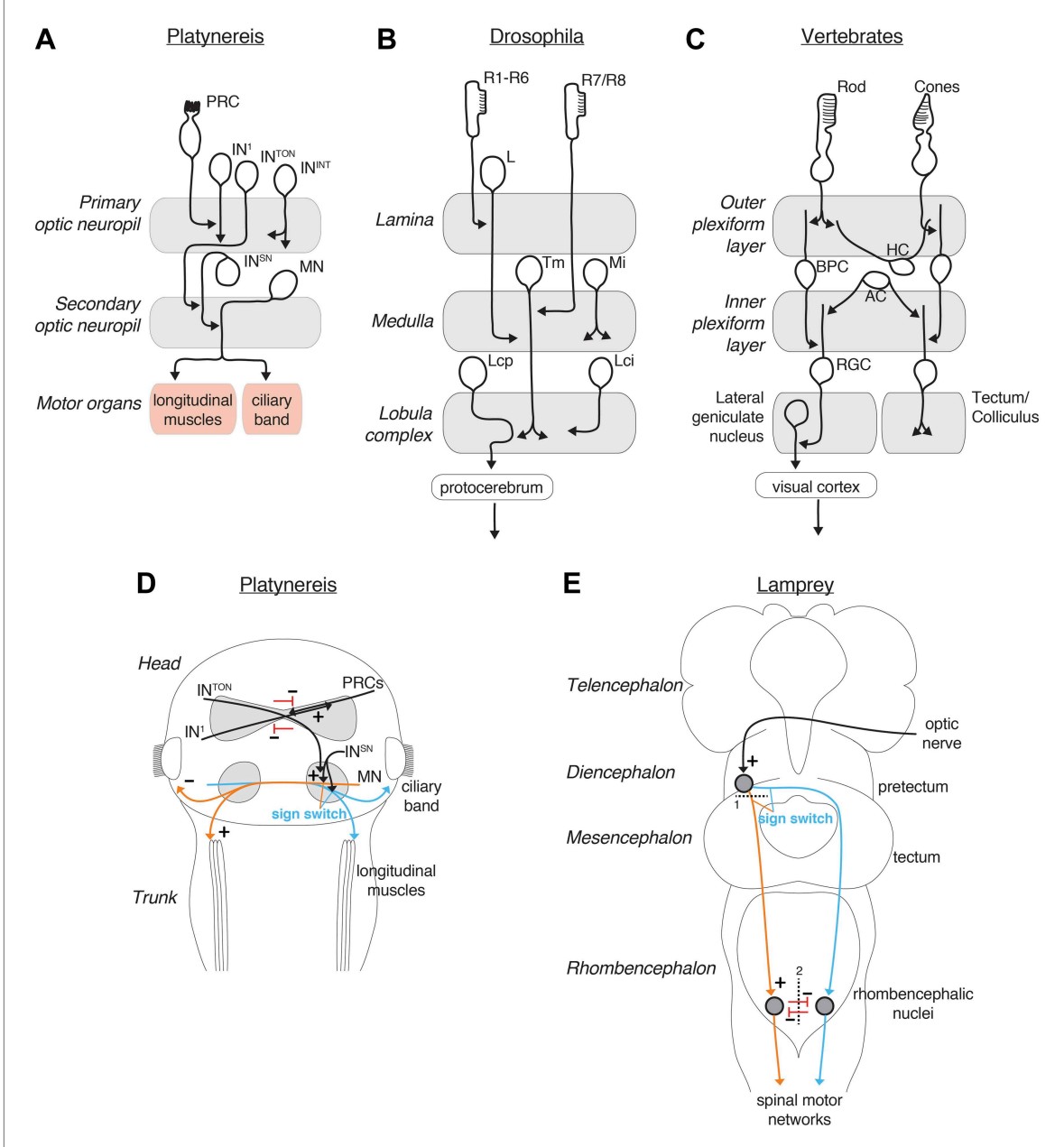

**Figure 8**. Comparison of the *Platynereis*, *Drosophila* and vertebrate visual circuits. Comparison of the *Platynereis* visual circuit with the *Drosophila* and vertebrate visual circuits on the neuronal level (**A–C**) and with the lamprey phototactic circuit on the circuit level (**D** and **E**). In (**E**) dashed line (1) represents a mesencephalic hemisection, severing connections between the pretectum and the ipsilateral reticulospinal neurons, dashed line (2) represents transection of the ventral rhombencephalic commissure. L, lamina monopolar neuron; Tm, transmedula neuron; Mi, medulla intrinsic neuron; Lcp, lobula complex projection neuron; Lci, lobula complex intrinsic neuron; BPC, bipolar cell; HC, horizontal cell, AC, amacrine cell; RGC, retinal ganglion cell. (**B** and **C**) after *Erclik et al. (2009)*; (*Sanes and Zipursky, 2010*) (**E**) after *Ullén et al. (1997)*.

with Formvar support film, contrasted with uranyl acetate and Reynold's lead citrate, and carbon coated to stabilize the film. Image acquisition of serial sections was performed at a pixel size of 3.72 nm on a FEI TECNAI Spirit transmission electron microscope (FEI, Hillsboro, Oregon) equipped with an UltraScan 4000 4 × 4k digital camera using the image acquisition software Digital Micrograph (Gatan, Pleasanton, CA). Stitching and alignment were accomplished using TrakEM2 (*Cardona et al., 2010*). The first 1245 layers were cut at 50 nm and aligned using an elastic alignment algorithm (*Saalfeld et al., 2012*). We later found that the larva was slightly tilted, and we did not reach the

muscles and the ciliary band on the right side of the animal. We therefore sectioned a further 445 sections at 40 nm, and aligned these using the rigid alignment algorithm in TrakEM2. All structures were segmented manually as area-lists or area-trees by an expert tracer (NR). Traces were exported into 3Dviewer, Imaris and Blender. Given the simple anatomy of the neurons and the large average diameter of axons tracing was in many cases unambiguous. Neurons with several short branches or with uncertain continuation were traced again by another expert tracer (LABC).

## Delineation of the eye circuit, neuron classification, nomenclature, and network visualization

The minimal eye circuit of the eyes contains only postsynaptic neurons downstream of photoreceptors. Neurons presynaptic to the minimal eye circuit at any level (with possible modulatory functions) are not included here. Five sensory cells, which are both pre- and post-synaptic to the minimal eye circuit were also excluded. 12 postsynaptic targets of the MNs in the ventral nerve cord and five between the secondary optic neuropils as well as 32 postsynaptic targets of interneurons were also excluded because of low connectivity and the occurrence of only individual synapses outside the optic neuropils.

48 neurite fragments could not be traced completely. Most of these fragments are short (<5 µm) with 1–5 synapses. They occur in regions where neurons have small short branches to receive or provide synapses. These fragments likely belong to identified neurons. Nine of the non-traceable fragments were longer (up to 44 µm).

We classified neurons according to their morphology, position, projection pattern and connectivity. Photoreceptors were identified based on the presence of a rhabdom located in the eye pigment cup. Their dominant targets are the four primary interneuron ($IN^1$) cells. $IN^1$ cells are named based on their position (anterior-left etc.). $IN^1$ cells are defined as a separate group based on their unique connectivity patterns, including strong input from photoreceptors and strong reciprocal contacts between the crosswise cells. The second group of interneurons is classified as $IN^{int}$s, with projections intrinsic to the primary optic neuropil and weak connectivity to photoreceptors and $IN^1$ cells. A third group is formed by the $IN^{ton}$ cells, distinguished by their unique projection pattern from the primary to the secondary optic neuropil. Schnörkel interneurons ($IN^{sn}$) form a distinct class with ventral cell bodies and curved axons ('Schnörkel' is German for 'curlicue') projecting to the ipsilateral secondary optic neuropil and synapsing on the motorneurons. The dorsal interneurons ($IN^{dc}$) have dorsally located cell bodies and axons that project to the secondary optic neuropil. The ventral interneurons ($IN^{vc}$) have ventrally located cell bodies and axons that project to the secondary optic neuropil. Motorneurons were recognized based on their innervation of dorsal and ventral longitudinal muscles as well as multi-ciliated cells (prototroch or metatroch). Motorneurons form two bilateral clusters. For some motorneurons we could not find motor synapses, however, we could classify these cells as motorneurons based on their cell body position and posteriorly projecting axons. Neurons of all types are labeled based on body side (left [l] or right [r]) and numbered (e.g., $IN^{tonl}1$).

Network analyses were done in Gephi 0.8.2. Modules were detected with an algorithm described in *Blondel et al. (2008)* with randomization on, using edge weights and a resolution of 1.2. Force-field-based clustering was performed using the Force Atlas 2 Plugin.

## Neuron visualization

Neurons and their connections reconstructed in TrakEM2 were exported as 3D objects in a wavefront (.obj) format and imported into Blender. Cell bodies were approximated with ellipsoids and axons and dendrites with smooth curves. The approximations were performed either manually or automatically using Python scripting embedded in Blender. Muscles were kept in the original form, pigment cups and ciliated cells were approximated with simplified 3D shapes based on TrakEM2 reconstructions. The model was extended with the connectivity and synapse position information obtained from TrakEM2.

## Virtual connectome atlas in blender

The virtual atlas (*Randel et al., 2014*) can be opened with Blender (http://www.blender.org/). Users new to Blender may consider watching a tutorial (http://www.blendtuts.com/2010/06/blender-25-interface.html). After opening the model file, two panels are visible in the *Tool shelf*: *Hide/Show groups* and *Highlight subnetworks* (Press 'T' if *Tool shelf* did not appear). If the panels do not appear, they can be loaded with the button *Reload Trusted* in the main menu panel (on the top). If menus still do not appear, the *Panel_show_group.py* and *Panel_synapses.py* python scripts need to be run by selecting the *Scripting* view in the main menu panel instead of *Default*. In the panel *Hide/Show groups* the cell types can be selected and hidden or displayed by pressing the *Apply* button. The panel

*Highlight subnetworks* enables querying pre- and post-synaptic cells by selecting a cell body of interest (right click) and pressing the *Postsynaptic* or *Presynaptic* buttons. The complete up- or downstream network of a cell can also be displayed by selecting a cell body and pressing the *Pre.subnetwork* or *Post.subnetwork* buttons. The *Show connectors* button visualizes the connectors between two cells after selecting the presynaptic cell first and then the postsynaptic cell. The *Show out connectors* button will reveal all outgoing synapses for a selected cell.

## Laser ablation

Laser ablation was performed on an Olympus FV1000 confocal microscope equipped with a SIM scanner (Olympus Corporation, Tokyo, Japan). Larvae were immobilized between a slide and a coverslip in NSW containing 100 mM MgCl$_2$. Larvae were imaged with an UPLSAPO 60X NA:1.20 water immersion objective using a 635-nm laser at 2–5% and transmission imaging with DIC optics. A 351-nm pulsed laser (Teem Photonics™, Grenoble, France) at 8–15% power was used, coupled via air to the SIM scanner for controlled ablation in a region of interest. 12% corresponds to a beam power of 168 µwatts as measured with a microscope slide power sensor (S170C; Thorlabs, Newton, NJ). During eye ablations we also imaged the eye pigment by reflection imaging of the 635-nm light using a PMT. Ablated larvae were placed into NSW for recovery (1–6 hr) before behavioral experiments.

## Selective eye illumination

Selective eye illumination was performed on an Olympus FV1000 confocal microscope equipped with a SIM scanner. Larvae were held by trapping them between a slide and a coverslip in natural seawater using several layers of tape as spacer. Larvae were imaged with an UPLSAPO 40X NA:0.90 air objective using a 635 nm laser at 1% and transmission imaging with DIC optics. The scanning speed was 2 µm/msec and we recorded 256 × 256 pixel images with a time increment of 0.43 s. For measuring the stimulus–response curve we used background illumination with the 488 nm laser at 10% power corresponding to 17.8 µwatts in the main scanner. For stimulations we used the 488 nm laser at 1–65% power via the SIM scanner, corresponding to a beam intensities of 0.53–20.9 µwatts. A circular region of interest of 25-pixel diameter (area of 490 pixels) covering the eyes was used for controlled illumination. The pixel dwell time of the beam was 10 µsec and the eyes were stimulated during 150 frames with the SIM scanner corresponding to a total exposure time of 735 msec during a 9.8 s trial. For the bending *Video 8* the stimulus laser power was at 10%. The laser power was measured with a microscope slide power sensor (S170C; Thorlabs). To record the spatial and temporal extent of the stimulus the reflected 488-nm light was imaged with a PMT. To prevent the detection of the 488-nm stimulus light by the transmitted light detector, we placed a red long-pass filter in front of the detector.

## In situ hybridization and image registration

Whole-mount in situ hybridization and image registration were performed as previously described (*Asadulina et al., 2012*). For probe generation we used expressed sequence tag clones or PCR-amplified fragments of already published *Platynereis* genes. The GeneBank accessions are: *TyrH*—JZ446954, *TrpH*–JZ446141, *VGlut*—JZ395359, *hdc*—JZ396646, *VAChT*—JZ402823, *ChAT*—JZ402100, *gad*—GU169427, dbh—KJ855061.

For registering gene expression patterns of 3-day-old larvae we generated a reference template (*Randel et al., 2014*) where we removed the bias present in the earlier reference, due to the use of a single larva as a starting template (*Asadulina et al., 2012*). In brief, we identified a median-size larva and registered it to every single image stack used for template generation (40 stacks), computed the average transformation and applied it to the median-size larva. The corrected larva was used as the starting point for the iteratively template generation (*Asadulina et al., 2012*). We registered minimum four individual larval whole-body scans for each gene to create an average expression pattern.

## Phototaxis assay

Larval phototaxis was assayed in a horizontal 6 × 15 mm rectangular glass cuvette with 3 mm high walls. The cuvette was illuminated uniformly through a light diffuser with white light from a 150-watt halogen cold light source (Schott KL 2500 LCD, Schott AG, Mainz, Germany). Larval behavior was recorded at 21 frames per second on a Zeiss Stemi 2000-CS stereomicroscope equipped with an AxioCam MRm camera (Carl Zeiss AG, Jena, Germany). Phototaxis efficiency was measured using a custom ImageJ macro that calculated larval density at the left and right side of the cuvette before and after the light stimulus (*Randel et al., 2014*).

## Calcium imaging

Fertilized eggs were injected with capped and polyA-tailed GCaMP6 (*Chen et al., 2013*) RNA generated from a vector (pUC57-T7-RPP2-GCaMP6) containing the GCaMP6 ORF fused to a 169 base pair 5′ UTR from the *Platynereis* 60S acidic ribosomal protein P2. 3-day-old larvae were held as described above. Imaging was performed on an Olympus FV1000 confocal microscope equipped with a SIM scanner. Larvae were imaged with an UPLSAPO 60X NA:1.20 water immersion objective using a 635-nm laser at 10% and transmission imaging with DIC optics. The GCaMP6 signal was imaged with a 488-nm laser at 2% intensity. The eyes were stimulated with the 488-nm laser in a region of interest using the SIM scanner. GCaMP6 signals were recorded simultaneously.

## Acknowledgements

We thank Dan Bumbarger and members of the Jékely group for comments on the manuscript, Ferdinand Kluge for help with tracing, Aurora Panzera for help with microscopy and microinjection, and Martin Gühmann for help with phototaxis scoring. The research leading to these results received funding from the European Research Council under the European Union's Seventh Framework Programme (FP7/2007-2013)/European Research Council Grant Agreement 260821.

Author contributions

GJ designed the research. NR prepared and sectioned the sample. RS imaged the sample. NR traced and assembled the connectome with help from AA and GJ. LABC and CV helped with the tracing and proofreading. NR and GJ performed laser ablations and behavioral experiments. EAW and MC performed gene expression analyses. AA visualized neuronal morphologies. GJ wrote the paper, with contributions from NR and AA.

## Additional information

### Funding

| Funder | Author |
|---|---|
| European Commission (EC) | Nadine Randel, Albina Asadulina, Elizabeth A Williams, Markus Conzelmann, Réza Shahidi |
| Max Planck Society | Luis A Bezares-Calderón, Csaba Verasztó, Gáspár Jékely |

The funders had no role in study design, data collection and interpretation, or the decision to submit the work for publication.

### Author contributions

NR, GJ, Conception and design, Acquisition of data, Analysis and interpretation of data, Drafting or revising the article; AA, Conception and design, Analysis and interpretation of data, Drafting or revising the article; LAB-C, CV, Analysis and interpretation of data, Drafting or revising the article; EAW, MC, RS, Acquisition of data, Analysis and interpretation of data, Drafting or revising the article

### Ethics

Animal experimentation: This study only used invertebrate animals.

## Additional files

### Major dataset

The following dataset was generated:

| Author(s) | Year | Dataset title | Dataset ID and/or URL | Database, license, and accessibility information |
|---|---|---|---|---|
| Randel N, Asadulina A, Bezares-Calderón LA, Verasztó C, Williams EA, Conzelmann M, Shahidi R, Jékely G | 2014 | Data from: Neuronal connectome of a sensory-motor circuit for visual navigation | doi: 10.5061/dryad.6f267 | Available at Dryad Digital Repository under a CC0 Public Domain Dedication. |

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
