## [Decision Letter]

Thank you for sending your work entitled “Neuronal connectome of a sensory-motor circuit for visual navigation” for consideration at *eLife*. Your article has been favorably evaluated by Eve Marder (Senior editor) and 3 reviewers.

The Senior editor and the other reviewers discussed their comments before we reached this decision, and the Senior editor has assembled the following comments to help you prepare a revised submission.

Your manuscript is obviously a unique one, and contains some invaluable work describing the connectome of a larval worm. This uniqueness is one of its most important strengths. Another strength is that it combines behavioral data with the EM data. That said, the manuscript requires considerable editorial revisions and potentially additional statistical analyses before it can be seen as suitable for publication. As seen below, parts of the data are described inadequately; there are missing statistics, and the connectivity diagrams are difficult to follow.

Major comments:

1) One of the reviewers was quite concerned about the reproducibility of the connectome as reported. Obviously, we can not expect you to do another complete set of sections and reconstructions. But it would be very instructive to know how variable different animals are. You report in Figure 5 data from a developmental series. This must come from multiple animals. First of all, please indicate the actual age of the animals at each data point. Secondly, for this one set of measurements, you should be able to look at a single age point and tell us how different the number of synapses are across animals? As it stands, you are not profiting well from these data, and they are impossible to interpret (no n's, no ages).

2) We understand that you saw no evidence of gap junctions. It is easy to miss these in EM, but have you specifically gone back to your images to look for them? Do you think there are likely to be there, but small, and therefore easily missed, or do you think, for any reason that they aren't there?

3) Not all synapses are the same. Do you see different classes of synaptic vesicles that might indicate some synapses are excitatory and some inhibitory?

4) It would also be useful to include figures showing the distribution of the synapses between the PRs and interneurons. For instance, it is not easy to figure out whether these axo-axonal synapses are clustered or distributed in the neuropils, and in particular, how their location correspond to ensheathing by surrounding glial cells. Spatial organization of synapses has important consequences on neuronal computation and it would also be very useful to determine how variable/invariant the synaptic organization is from animal to animal, or even from PR-IN to PR-IN connections within the same animal.

5) The inadequacy of the circuit diagrams is best expressed by this comment from one reviewer. Please think about creating more useful connectivity diagrams. "This is a difficult paper to assess. On the positive side, it carefully and fully documents a very nice piece of research on the larval visual center and visual response of Platynereis, an interesting and promising model system, using up-to-date methodologies. For the specialist, it provides a thorough summary of the data and, through the Blender software (which I have not tried), I gather there is the option to access the data directly. On the negative side, for a general reader like myself, there is the frustrating sense that the results have not been fully developed into a set of meaningful conclusions, or at least a set of hypotheses, for comparison with results on other types of visual circuits, e.g., in vertebrate retina or Drosophila. The conclusions in a neuroscience paper can typically be summarized in one or more circuit diagrams showing the cell types in a schematic way, with their synaptic and non-synaptic contacts, and indications of likely + or - effects, so the logic of the circuit can be at least provisionally discussed in a concrete way. The current paper instead adopts a connectome approach and language, which is essentially statistical and in many ways more nebulous. This is entirely appropriate for dealing with data from really complex systems, where a circuitry approach is beyond reach. For a comparatively simple system like Platynereis, one would hope the circuitry approach would provide sufficient insight to be worthwhile, and might yield a simple take-home message or two.”

6) Many of the behavioral results are presented without the appropriate stat and tests indicated. You may have omitted these because the results are so obvious, but nonetheless, each figure legend minimally should have in it the appropriate statistical tests.

7) Please address the following comments:

Figure 1 legend: Since this is the first figure, and acts as a guide to the structure, the main abbreviations should be introduced, even though this duplicates the text. So, what are PRCs, INs and MNs (details, e.g., superscripts, can be left out here), and it looks like the colors are consistent in D-F, i.e., two shades of blue for the PRCs and green, pink and yellow for the INs, which is helpful to know. The yellow barbell in H is not explained, and obscures the vesicles, which are not very distinct in this example in any case.

In the Results section, the reference to Tbars being absent; this seems to be either mis-stated or misplaced in the sentence so as to be ambiguous. Arthropod Tbars are presynaptic, but here it's the post-synaptic density that is being referred to. As an aside, I wondered about the indistinctness of the synapses generally, in Figure 1 and the supplementary examples. Perhaps a further note is required on the fixation, staining and imaging, which seems not to give very clear membrane profiles. Annelid synapses do not have very marked post-synaptic densities in my experience, but these appear even fainter than usual, so maybe more needs to be said about the criteria for assessing them.

Again, in the Results section, I assume it's the photoreceptor axons that cross the midline. Also, in relation to the main point made above, I realize the use throughout of phrases like “connecting with...” is part of the connectome terminology. But more precise conventional terms, e.g., “one-way (or mutual) synaptic contacts between...” conveys more information to someone reading the account from beginning to end for information. There is, in short, a cost to this type of presentation.

Figure 4: This is clearly the closest the authors are going to get to a circuit diagram, but I confess to being rather vague on the analytical methods and terminology, so the precise meaning, e.g., of the thickness of connecting lines, is not immediately obvious to me. However, in general it looks like strong redundancy in the upstream connections and interactions between the four primary interneurons. In contrast, connections elsewhere are diffuse and far less redundant. To me this immediately suggests there should be command nexus, i.e., a set of (perhaps not morphologically very obvious) key command synapses linking the primary INs to the rest of the circuit. If not, it would imply the downstream behavioral response should be both labile and very finely graded, i.e., subject to numerous competing secondary modulations of equal strength, which may or may not correlate with what the behavioral data is showing. A proper circuit analysis might make the alternatives clearer, without requiring readers to go back to the raw data and immerse themselves in it.

General discussion: Transmitters are not mentioned, yet there is a fair amount of published information on the molecular identity of Platynereis neurons. In part, because these previous papers also avoid summary diagrams, it is difficult for a non-specialist to compare those results with the cells described here. Are the some or all of the cells described here ones for which the transmitter is not known, or has this been determined? Knowing the transmitters, and even just the vesicle type, one should have some idea of the nature of the synapses, e.g., +, - or modulatory, and this would contribute to the reader's grasp of system function. For example, since histamine is the PRC transmitter in arthropods, is it the same in annelids or not – if the former then this tells us something about the likely nature of the PRC synapses in this system, which should be useful.

---

## [Author Response]

*1) One of the reviewers was quite concerned about the reproducibility of the connectome as reported. Obviously, we can not expect you to do another complete set of sections and reconstructions. But it would be very instructive to know how variable different animals are. You report in*
Figure 5
*data from a developmental series. This must come from multiple animals. First of all, please indicate the actual age of the animals at each data point. Secondly, for this one set of measurements, you should be able to look at a single age point and tell us how different the number of synapses are across animals? As it stands, you are not profiting well from these data, and they are impossible to interpret (no n's, no ages)*.

To address this concern we have now included an additional section “Stereotypy of synaptic connections”. Given that all our reconstructions were done from a single individual, we cannot yet address the variability of synaptic connections between individual larvae. However, we have compared the connectivity matrices of the left and right sides of the body in our ssTEM larval sample and found that they are significantly correlated. We also analyzed the distribution of synapses on left and right neurons of the same type. We found that the spatial distribution of synapses is stereotypical for the same neuron types (e.g., Figure 4—figure supplement 5).

The data about the development of photoreceptor connections come from the same 3-day old individual. Since photoreceptors are added continuously to the eye during development, we could analyze the connectivity of photoreceptors at different states of cellular differentiation. We explain this more clearly in the revised text.

2) We understand that you saw no evidence of gap junctions. It is easy to miss these in EM, but have you specifically gone back to your images to look for them? Do you think there are likely to be there, but small, and therefore easily missed, or do you think, for any reason that they aren't there?

We have now compiled and provided a set of high-resolution images (1.13 nm/pixel resolution) focusing on part of the primary optic neuropil. We performed a comprehensive search of these images for structures with similar to annelid gap junctions as described in the literature, however, no structures reliably identifiable as gap junctions were present. Although transcriptomic studies indicate that Platynereis has several innexin genes, gap junctions may be rare or too small in the larval stage to be identified, even in high resolution scans (not feasible for large-scale connectomics).

3) Not all synapses are the same. Do you see different classes of synaptic vesicles that might indicate some synapses are excitatory and some inhibitory?

We have now included an additional section on synapse types “Synapse types in the eye circuitry”. Using high-resolution scans from several synapses from different neuron types we have quantified the diameter of synaptic vesicles and found significantly different mean diameters. To characterize the neurotransmitters of the eye circuit we also included *in situ* hybridization data in 3-day-old larvae for markers genes of all major neurotransmitters. Using image registration and double *in situ* hybridization, we could determine the transmitter type for the photoreceptors (glutamate), interneurons (different monoamines), and motorneurons (acetylcholine/GABA).

*4) It would also be useful to include figures showing the distribution of the synapses between the PRs and interneurons. For instance, it is not easy to figure out whether these axo-axonal synapses are clustered or distributed in the neuropils, and in particular, how their location correspond to ensheathing by surrounding glial cells. Spatial organization of synapses has important consequences on neuronal computation and it would also be very useful to determine how variable/invariant the synaptic organization is from animal to animal, or even from PR-IN to PR-IN connections within the same animal*.

We have now included a new figure (Figure 4—figure supplement 5) and a video file (Video 4) showing the distribution of PRC to interneuron synapses and how these synapses are positioned relative to the synapses between the reciprocally connected crosswise primary interneurons.

*5) The inadequacy of the circuit diagrams is best expressed by this comment from one reviewer. Please think about creating more useful connectivity diagrams. “This is a difficult paper to assess. On the positive side, it carefully and fully documents a very nice piece of research on the larval visual center and visual response of Platynereis, an interesting and promising model system, using up-to-date methodologies. For the specialist, it provides a thorough summary of the data and, through the Blender software (which I have not tried), I gather there is the option to access the data directly. On the negative side, for a general reader like myself, there is the frustrating sense that the results have not been fully developed into a set of meaningful conclusions, or at least a set of hypotheses, for comparison with results on other types of visual circuits, e.g., in vertebrate retina or Drosophila. The conclusions in a neuroscience paper can typically be summarized in one or more circuit diagrams showing the cell types in a schematic way, with their synaptic and non-synaptic contacts, and indications of likely + or - effects, so the logic of the circuit can be at least provisionally discussed in a concrete way. The current paper instead adopts a connectome approach and language, which is essentially statistical and in many ways more nebulous. This is entirely appropriate for dealing with data from really complex systems, where a circuitry approach is beyond reach. For a comparatively simple system like Platynereis, one would hope the circuitry approach would provide sufficient insight to be worthwhile, and might yield a simple take-home message or two*.*”*

We have changed the layout of several of the circuit diagrams to better represent the logic of the *Platynereis* larval eye circuit. We have also included a new figure (Figure 8) showing a schematic circuit diagram and compare it to the phototactic circuit of the lamprey. We also compare the organization of the visual neuropils in *Platynereis* to that in *Drosophila* and vertebrate. We also altered the text in several places and include a more explicit discussion of the circuit in the Discussion. We think, however, that a connectomics approach and language is also warranted, and in fact helped us to discover important features of the circuit, for example the crosswise, point symmetric and reciprocally connected modules of the four eyes and their interneurons.

*6) Many of the behavioral results are presented without the appropriate stat and tests indicated. You may have omitted these because the results are so obvious, but nonetheless, each figure legend minimally should have in it the appropriate statistical tests*.

We have now included all statistical tests performed in both the figures and the figure legends.

7) Please address the following comments:

Figure 1
*legend: Since this is the first figure, and acts as a guide to the structure, the main abbreviations should be introduced, even though this duplicates the text. So, what are PRCs, INs and MNs (details, e.g., superscripts, can be left out here), and it looks like the colors are consistent in D-F, i.e., two shades of blue for the PRCs and green, pink and yellow for the INs, which is helpful to know. The yellow barbell in H is not explained, and obscures the vesicles, which are not very distinct in this example in any case*.

Corrected.

*In the Results section, the reference to Tbars being absent; this seems to be either mis-stated or misplaced in the sentence so as to be ambiguous. Arthropod Tbars are presynaptic, but here it's the post-synaptic density that is being referred to. As an aside, I wondered about the indistinctness of the synapses generally, in*
Figure 1
*and the supplementary examples. Perhaps a further note is required on the fixation, staining and imaging, which seems not to give very clear membrane profiles. Annelid synapses do not have very marked post-synaptic densities in my experience, but these appear even fainter than usual, so maybe more needs to be said about the criteria for assessing them*.

We corrected the sentence referring to Tbars. We generated high-resolution images of 60 random selected synapses distributed across all neuron types. We have adjusted the figure showing the synapses and have now included several examples at low- (3.7 nm/pixel, used for tracing) and high-resolution (0.2. nm/pixel, used to characterize synapses in detail). We found an accumulation of vesicles close to the membrane at all sites that were identified as synapses in the low-resolution scans. We also quantified synaptic vesicle diameter for several synapses. We have defined the criteria for synapse identification in the “Morphology of neurons and synapses” section. We provided a detailed description of the fixation and contrasting protocol in the Methods section.

*Again, in the Results section, I assume it's the photoreceptor axons that cross the midline*.

Corrected.

*Also, in relation to the main point made above, I realize the use throughout of phrases like “connecting with...” is part of the connectome terminology. But more precise conventional terms, e.g., “one-way (or mutual) synaptic contacts between...” conveys more information to someone reading the account from beginning to end for information. There is, in short, a cost to this type of presentation*.

We have now altered the text at several places and use more conventional circuit terminology, with the exception of the sections discussing the graph analyses.

Figure 4*: This is clearly the closest the authors are going to get to a circuit diagram, but I confess to being rather vague on the analytical methods and terminology, so the precise meaning, e.g., of the thickness of connecting lines, is not immediately obvious to me. However, in general it looks like strong redundancy in the upstream connections and interactions between the four primary interneurons. In contrast, connections elsewhere are diffuse and far less redundant. To me this immediately suggests there should be command nexus, i.e., a set of (perhaps not morphologically very obvious) key command synapses linking the primary INs to the rest of the circuit. If not, it would imply the downstream behavioral response should be both labile and very finely graded, i.e., subject to numerous competing secondary modulations of equal strength, which may or may not correlate with what the behavioral data is showing. A proper circuit analysis might make the alternatives clearer, without requiring readers to go back to the raw data and immerse themselves in it*.

We now explain in the figure legend that the thickness of the connecting lines is proportional to the number of synapses between two neurons. We have also included further discussion on the logic of the circuit, and how it compares to the lamprey phototactic circuit, in the Discussion. We included an extra figure (Figure 8) comparing the insect, vertebrate and *Platynereis* visual circuits.

*General discussion: Transmitters are not mentioned, yet there is a fair amount of published information on the molecular identity of Platynereis neurons. In part, because these previous papers also avoid summary diagrams, it is difficult for a non-specialist to compare those results with the cells described here. Are the some or all of the cells described here ones for which the transmitter is not known, or has this been determined? Knowing the transmitters, and even just the vesicle type, one should have some idea of the nature of the synapses, e.g., +, - or modulatory, and this would contribute to the reader's grasp of system function. For example, since histamine is the PRC transmitter in arthropods, is it the same in annelids or not – if the former then this tells us something about the likely nature of the PRC synapses in this system, which should be useful*.

We now include an extensive analysis of the neurotransmitters (Figure 3 and associated supplementary figures, and source data, and also see reply to major comment 3).